# Benchmarking Counterfactual Image Generation

**Thomas Melistas**[*,1,2,3], **Nikos Spyrou**[*,1,2,3], **Nefeli Gkouti**[*,1,2,3], **Pedro Sanchez**[3], **Athanasios Vlontzos**[4,6], **Yannis Panagakis**[1,2], **Giorgos Papanastasiou**[2,5], **Sotirios A. Tsaftaris**[2,3]

[1]National & Kapodistrian University of Athens, Greece
[2]Archimedes/Athena RC, Greece
[3]The University of Edinburgh, UK
[4]Imperial College London, UK
[5]The University of Essex, UK
[6]Spotify
`https://gulnazaki.github.io/counterfactual-benchmark/`

## Abstract

Generative AI has revolutionised visual content editing, empowering users to effortlessly modify images and videos. However, not all edits are equal. To perform realistic edits in domains such as natural image or medical imaging, modifications must respect causal relationships inherent to the data generation process. Such image editing falls into the counterfactual image generation regime. Evaluating counterfactual image generation is substantially complex: not only it lacks observable ground truths, but also requires adherence to causal constraints. Although several counterfactual image generation methods and evaluation metrics exist, a comprehensive comparison within a unified setting is lacking. We present a comparison framework to thoroughly benchmark counterfactual image generation methods. We evaluate the performance of three conditional image generation model families developed within the Structural Causal Model (SCM) framework. We incorporate several metrics that assess diverse aspects of counterfactuals, such as composition, effectiveness, minimality of interventions, and image realism. We integrate all models that have been used for the task at hand and expand them to novel datasets and causal graphs, demonstrating the superiority of Hierarchical VAEs across most datasets and metrics. Our framework is implemented in a user-friendly Python package that can be extended to incorporate additional SCMs, causal methods, generative models, and datasets for the community to build on. Code: `https://github.com/gulnazaki/counterfactual-benchmark`.

## 1 Introduction

Generative AI has revolutionized visual content editing, empowering users to effortlessly modify images and videos (9; 30; 45; 34; 17). However, not all edits are equal, especially in complex domains such as natural or medical imaging, where image realism is paramount (58; 23). In these fields, modifications must adhere to the underlying causal relationships that govern the data generation process. Ignoring these, can lead to unrealistic and potentially misleading results, undermining the integrity of the edited content (46; 58). Image editing inherently relies on *implied* causal relationships. However, learned generative models powering image editing, will not explicitly account for these causal relations and can produce unrealistic results. Figure 1 showcases clearly what happens when causal relationships are not taken into account and motivates the need for causally faithful approaches.

---

[*]These authors contributed equally to this work

38th Conference on Neural Information Processing Systems (NeurIPS 2024) Track on Datasets and Benchmarks.

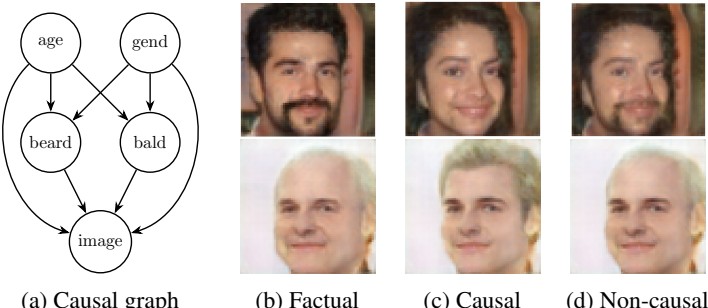

| (a) Causal graph | (b) Factual | (c) Causal | (d) Non-causal |

Figure 1: (a) A plausible causal graph for human faces; (b) Factual images (no intervention); (c) Causal counterfactual images using the graph of (a) to perform the interventions $do(Female)$ (upper panel) and $do(Young)$ (lower panel); (d) Non-causal image editing.

Image editing, where modifications intentionally alter the underlying causal relationships within the image to explore hypothetical "what if" scenarios, falls squarely into the counterfactual image generation regime (46). This emerging field seeks to generate images that depict plausible yet alternative versions of reality, allowing us to visualize the potential consequences of changes to specific attributes or conditions. Figure 1c displays image edits that take into account the causal graph of Figure 1a, while Figure 1d depicts image edits that do not. Intervening on Gender (*do(Female)*) results in a female person with a beard, while intervening on Age (*do(Young)*) results in a young person with baldness, which do not account for the causal relationships in the data-generation process.

Evaluating counterfactual image generation is substantially challenging (62; 46). Not only do these hypothetical scenarios lack observable ground truths, but their assessment requires adhering to causal constraints whilst ensuring the generated images are of high quality and visually plausible. This multifaceted problem demands a nuanced approach that balances the need for realism with the constraints imposed by causal relationships.

Under the Pearlian framework, we adopt a set of Structural Causal Models (SCM) (38) to explicitly inform generative models about causal paths between high-level variables. Based on the SCM, we employ the counterfactual inference paradigm of *Abduction-Action-Prediction* and the mechanisms of Deep Structural Causal Models (Deep-SCM) (36) to evaluate the performance of various methods. As such, in this paper, we introduce a comprehensive framework, able to extensively evaluate any published methods on SCM-based counterfactual image generation. We aspire for this work to become the *de facto benchmark* for the community when developing new methods (and metrics) in counterfactual image generation.

Briefly, the main **contributions** of our work are: **(1)** We develop a comprehensive framework to evaluate the performance of *all* image generation models published under the Deep-SCM paradigm. We explore several datasets (synthetic, natural and medical images), as well as SCM structures across datasets. **(2)** We expand the above models to accommodate previously untested datasets and causal graphs. Specifically, we test HVAE and GAN on a non-trivial causal graph for human faces and we devise a GAN architecture inspired from (59) that can generate counterfactual brain MRIs given multiple variables. **(3)** We extensively benchmark these models adopting several metrics to evaluate causal SCM-based counterfactual image generation. **(4)** We offer a user-friendly Python package to accommodate and evaluate forthcoming causal mechanisms, datasets and causal graphs.

## 2 Related work

**Causal Counterfactual Image Generation:** Pairing SCMs with deep learning mechanisms for counterfactual image generation can be traced back to the emergence of Deep-SCM (36). The authors utilised Normalising Flows (51) and variational inference (25) to infer the exogenous noise and perform causal inference under the *no unobserved confounders* assumption. Follow-up work (7) incorporates a Hierarchical VAE (HVAE) (5) to improve the fidelity of counterfactuals. In parallel, Generative Adversarial Networks (GANs) (14; 10) have been used to perform counterfactual inference through an adversarial objective (6) and for interventional inference, by predicting reparametrised distributions over image attributes (28). Variational Autoencoders (VAEs) have been used for

counterfactual inference (61), focusing on learning structured representations from data and capturing causal relationships between high-level variables, thus diverging from the above scope. Diffusion models (21; 48), on the other hand, were also recently used to learn causal paths between high-level variables (44; 43; 12), approaching the noise abduction as a forward diffusion process. However, these works (44; 43; 12) address the scenario of a single variable (attribute) affecting the image.

Similarly to Deep-SCM, Normalising Flows and VAEs were used in (27), following a backtracking approach (29). The aforementioned method involves tracing back through the causal graph to identify changes that would lead to the desired change in the attribute intervened. Another line of work, (42), utilises deep twin networks (57) to perform counterfactual inference in the latent space, without following the *Abduction-Action-Prediction* paradigm. Since these methods deviate from our scope and focus on Deep-SCM-based causal counterfactuals, we chose not to include them in the current benchmark. As we aim to explore further causal mechanisms, models and data, we believe that these methods are worthwhile future extensions of our current work.

Recent works based on diffusion models (21) have achieved SoTA results on image editing (22; 34; 17; 2; 60; 54), usually by altering a text prompt to edit a given image. While the produced images are of high quality, we aim to explicitly model the causal interplay of known high-level variables, an approach not yet applied to diffusion models to the best of our knowledge. Therefore, even though the task of image editing exhibits similarities to counterfactual image generation, we focus on true causal counterfactual methods, as we discussed above.

**Evaluation of counterfactuals:** To the best of our knowledge, there is no prior work offering a comprehensive framework to thoroughly evaluate the performance, fidelity, and reliability of counterfactual image generation methods, considering both image quality and their relationship to factual images and intervened variables. This paper aims to address this gap. A study relevant to ours is (52): it compares established methods for counterfactual explainability[1] (8; 32). However, its main focus lies on creating counterfactual explanations for classifiers on a single dataset (MorphoMNIST (4)) using solely two causal generative models (VAE, GAN) (36; 6).

Various metrics have been proposed to evaluate counterfactual image generation. For instance, Monteiro et al. (35) introduce metrics, following an axiomatic definition of counterfactuals (15; 13), namely the properties that arise from the mathematical formulation of SCMs. The authors in (56) evaluate sparsity of counterfactual explanations via elastic net loss and similarity of factual versus counterfactual distributions via autoencoder reconstruction errors. Sanchez and Tsaftaris (44) introduce a metric to evaluate minimality in the latent space. In this work, we adopt the metrics of (35) and (44).

Optimising criteria of causal faithfulness as those above does not directly ascertain image quality. Image quality evaluation metrics used broadly in the context of image editing include the Fréchet inception distance (FID), (19) and similarity metrics such as the Learned Perceptual Image Patch Similarity (LPIPS) (63) and CLIPscore (18). We adopt such image quality evaluation metrics herein.

## 3 Methodology

### 3.1 Preliminaries

**SCM-based counterfactuals** The methods we are comparing fall into the common paradigm of SCM-based interventional counterfactuals (37). A Structural Causal Model (SCM) $\mathcal{G} := (\mathbf{S}, P(\boldsymbol{\epsilon}))$ consists of: **(i)** A collection of structural assignments, called mechanisms $\mathbf{S} = \{f_i\}_{i=1}^{N}$, s.t. $x_i = f_i(\epsilon_i, \mathbf{pa}_i)$; and **(ii)** A joint distribution $P(\boldsymbol{\epsilon}) = \prod_{i=1}^{N} p(\epsilon_i)$ over mutually independent noise variables, where $x_i$ is a random variable, $\mathbf{pa}_i$ are the parents of $x_i$ (its direct causes) and $\epsilon_i$ is a random variable (noise).

Causal relations are represented by a directed acyclic graph (DAG). Due to the acyclicity, we can recursively solve for $x_i$ and obtain a function $\mathbf{x} = \mathbf{f}(\boldsymbol{\epsilon})$. We assume $\mathbf{x}$ to be a collection of observable variables, where $x_i$ can be a high-dimensional object such as an image and $x_{j \neq i}$ image attributes. The variables $\mathbf{x}$ are observable, hence termed *endogenous*, while $\boldsymbol{\epsilon}$ are unobservable, *exogenous*.

---

[1]Counterfactual explanations aim to explain changes in the predictions of classifiers through minimal changes of the input. Although both our task and counterfactual explainability can benefit from each other, they differ in terms of their objective and methodology.

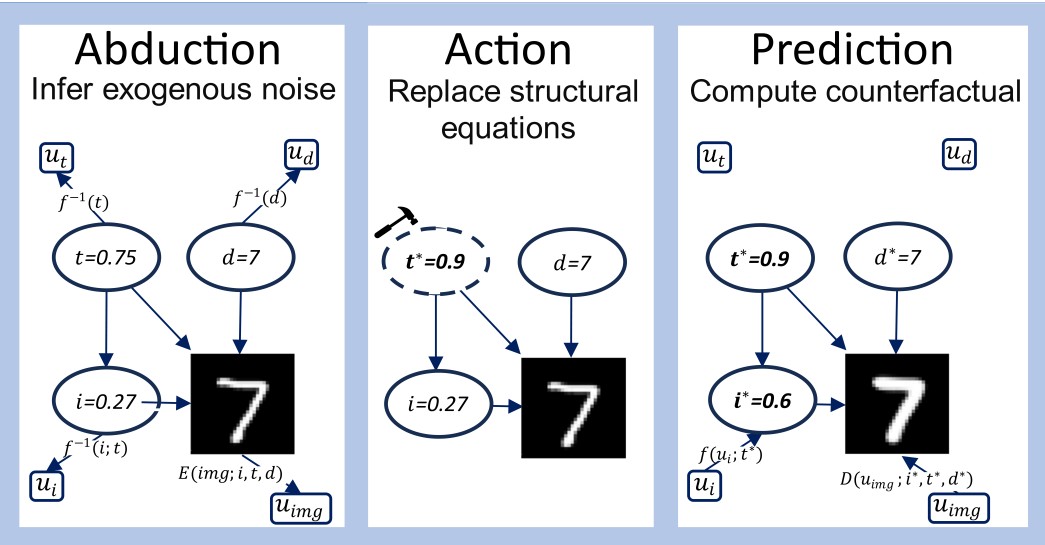

Figure 2: **Inference example on MorphoMNIST:** *Abduction*: We abduct the exogenous noise separately for each variable, using inverse Normalising Flows $f^{-1}$ for the attributes and the encoder of the image mechanism (e.g. VAE, HVAE or GAN) for the image, conditioning on the factual parents. *Action*: We intervene $(do(t^*))$ only on *thickness*. *Prediction*: We employ the Normalizing Flow $f$ conditioned on the counterfactual thickness $t^*$ to obtain $i^*$ after the intervention. Note that this is not needed for $t^*$ on which we intervene and for $d^*$ that has no parents in the DAG. Finally, the decoder generates the counterfactual image, given the exogenous noise $U_{img}$ and conditioning on the counterfactual parents.

The models we examine make several assumptions: **(i)** Known causal graph (captured by a SCM). **(ii)** Lack of unobserved confounders (causal sufficiency), making the $\epsilon_i$, $\epsilon_j$ mutually independent. **(iii)** Invertible causal mechanisms, i.e. $\epsilon_i = f_i^{-1}(x_i, \mathbf{pa}_i) \; \forall \, i \in [1, n]$ and hence $\boldsymbol{\epsilon} = \mathbf{f}^{-1}(\mathbf{x})$.

Thanks to the causal interpretation of SCMs, we can compute *interventional* distributions, namely predict the effect of an *intervention* to a specific variable, $P(x_j | do(x_i = y))$. Interventions are formulated by substituting a structural assignment $x_i = f_i(\epsilon_i, \mathbf{pa}_i)$ with $x_i = y$. Interventions operate at the population level, so the unobserved noise variables are sampled from the prior $P(\boldsymbol{\epsilon})$. *Counterfactual* distributions instead refer to a specific observation and can be written as $P(x_{j, x_i = y} | \mathbf{x})$. We assume that the structural assignments change, as previously, but the exogenous noise is identical to the one that produced the observation. For this reason, we have to compute the posterior noise $P(\boldsymbol{\epsilon} | \mathbf{x})$. Counterfactual queries, hence, can be formulated as a three-step procedure, known as the *Abduction-Action-Prediction* paradigm: **(i)** *Abduction*: Infer $P(\boldsymbol{\epsilon} | \mathbf{x})$, the state of the world (exogenous noise) that is compatible with the observation $\mathbf{x}$. **(ii)** *Action*: Replace the structural equations $do(x_i = y)$ corresponding to the intervention, resulting in a modified SCM $\widetilde{\mathcal{G}} := \mathcal{G}_{\mathbf{x}; do(x_i = y)} = (\widetilde{\mathbf{S}}, P(\boldsymbol{\epsilon} | \mathbf{x}))$. **(iii)** *Prediction*: Use the modified model to compute $P_{\widetilde{\mathcal{G}}}(\mathbf{x})$.

## 3.2 SCM Mechanisms and Models

Counterfactual image generation examines the effect of a change to a parent variable (attribute) on the image. To enable the *Abduction-Action-Prediction* paradigm we consider three categories of invertible deep learning mechanisms and the corresponding models that utilise them: **(1)** *Invertible, explicit* implemented with *Conditional Normalising Flows* (53); **(2)** *Amortized, explicit* utilised with *Conditional VAEs* (25; 20) or extensions such as *Hierarchical VAEs* (26; 47); **(3)** *Amortized, implicit* (6), implemented with *Conditional GANs* (33; 10). The first mechanism is invertible by design and is employed for the attribute mechanisms, while the latter two are suitable for high-dimensional variables such as images and achieve invertibility through an amortized variational inference and an adversarial objective, respectively.

For each variable of the SCM examined, we train a model independently to perform noise abduction, considering conditional Normalising Flows for the parent attributes and conditional VAEs, HVAEs and GANs for the image. These model families encompass all the aforementioned types of invertible mechanisms, as discussed by Pawlowski et al. (36). Further details and background of these methods and particularly how they enable the *Abduction-Action-Prediction* paradigm can be found in Appendix A.1. During counterfactual inference, all the posterior noise variables (given the factual) are abducted independently, except for the intervened variables that are replaced with a constant. Respecting the order of the causal graph, we infer all variables, given the abducted noise, according to the modified structural assignments. The inference procedure is visualised in Figure 2.

Finally, following the formulation in Appendix A.1, we must note that for VAEs and HVAEs the counterfactual image $x^*$ is inferred as an affine transformation of the factual $x$, [2] while for GANs all the information for the factual is passed through the latent.

### 3.3 Evaluation Metrics

The evaluation of counterfactual inference has been formalised through the axioms of (13; 15). Monteiro et al. (35) utilise such an axiomatic definition to introduce three metrics to evaluate image counterfactuals: *Composition*, *Effectiveness*, and *Reversibility*.[3] While it is necessary for counterfactual images to respect these axiom-based metrics, we find that they are not sufficient for a perceptual definition of successful counterfactuals. The additional desiderata we consider are the *realism* of produced images, as well as the *minimality* (or sparseness) of any changes made. All adopted metrics **do not** require access to ground truth counterfactuals. We now briefly describe them, leaving detailed formulation and implementation details in the Appendix A.2.

***Composition*** guarantees that the image and its attributes do not change when performing a *null-intervention*, namely when we skip the *action* step (35). It measures the ability of the mechanisms to reproduce the original image. We perform the *null-intervention* $m$ times and measure the distance to the original image. We use the $l_1$ distance on image space, as well as the LPIPS distance (63).

***Effectiveness*** determines if the intervention was successful. In order to quantitatively evaluate effectiveness for a given counterfactual image we leverage anti-causal predictors trained on the observed data distribution, for each parent variable $pa_x^i$ (35). Each predictor, then, approximates the counterfactual parent $pa_x^{i*}$ given the counterfactual image $x^*$ as input. We employ classification metrics (accuracy, F1) for categorical variables and regression metrics (mean absolute error) for continuous.

***Realism*** measures counterfactual image quality by capturing its similarity to the factual. To evaluate realism quantitatively, we employ the *Fréchet Inception Distance* (FID) (19)

***Minimality*** evaluates whether the counterfactual only differs according to the modified parent attribute against the factual, ideally leaving all other attributes unaffected. While we can achieve an approximate estimate of minimality by combining composition and effectiveness, an additional metric is helpful to measure proximity to the factual. For this reason, we leverage the *Counterfactual Latent Divergence* (CLD) metric introduced in (44). CLD calculates the "distance" between the counterfactual and factual images in a latent space. Intuitively, CLD represents a trade-off, ensuring the counterfactual is sufficiently distant from the factual class, but not as far as other real images from the counterfactual class are.

## 4 Results

**Datasets** Given the availability of architectures and demonstrated outcomes in the literature, we benchmark all the above models on the following three datasets: **(i)** MorphoMNIST (4), **(ii)** CelebA (31) and **(iii)** ADNI (39). MorphoMNIST is a purely synthetic dataset generated by inducing morphological operations on MNIST with a resolution of 32×32. We note, that interventions on thickness affect both the intensity and image (Figure 3(a)). For CelebA (64×64) we use both a simple

---

[2]Formally, $x^* = (\sigma^* \oslash \sigma \odot x + \mu^* - \sigma^* \oslash \sigma \odot \mu)$, given networks $\mu := \mu_\theta(z, pa_x)$ and $\sigma := \sigma_\theta(z, pa_x)$ generating per pixel mean/std outputs given noise $z$ and parents $pa_x$. $\odot, \oslash$: per-pixel multiplication, division.

[3]We did not include reversibility, since the applied interventions cannot always be cycle-consistent. This was also the case in the follow up work by the same authors (7).

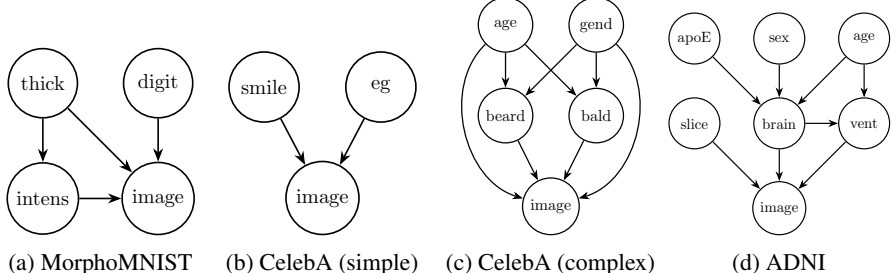

|       |       |       |       |
| ----- | ----- | ----- | ----- |
| (a) MorphoMNIST | (b) CelebA (simple) | (c) CelebA (complex) | (d) ADNI |

Figure 3: Causal graphs for all examined datasets.

graph with two assumed disentangled attributes as done in (35) (Figure 3(b)), as well as we devise a more complex graph that introduces attribute relations taken from (27) (Figure 3(c)). *Alzheimer's Disease Neuroimaging Initiative* (ADNI) provides a dataset of brain MRIs from which we use the *ADNI1 standardised* set (all MRIs acquired at 1.5T). The causal graph we use (Figure 3(d)) is inspired by (1) from which we removed all missing variables (See Appendix B.1 for more details).

**Setup** For Normalising Flows we base our implementation on (27), using Masked Affine Autoregressive (11), Quadratic Spline (11) and ScaleShift flows, as well as utilising the Gumbel-max parametrisation for discrete variables as done in (36). For **MorphoMNIST**, we compare (i) the VAE architecture as given in the original Deep-SCM paper (36), (ii) the HVAE architecture of (7) and (iii) the fine-tuned GAN architecture of (52) as the set up of (6) did not converge. For **CelebA**, we compare (i) the VAE architecture proposed as a baseline in (6), (ii) the GAN architecture of (6), as well as (iii) the HVAE used in (35). We further test this setup with a complex graph examined in (27), using it to assess how the former GAN and HVAE architectures behave in previously untested, more challenging scenarios. Finally, for brain MRIs (**ADNI**), we (i) adjust the VAE architecture of (1) to generate higher resolution images and (ii) use the HVAE that was developed in (7) for another dataset of brain MRIs (UK Biobank (49)). Simple adjustments to the GAN architecture of (6) would not lead to convergence, even with rigorous parameter tuning. Therefore, we resorted to the architecture of (59), which we modified by removing the residual connections between the encoder and the decoder to adhere to the *amortized, implicit* setup.

For all datasets and graphs we trained the VAE model with learnable standard deviation, as used in (7), with the formulation provided in Appendix A.1. We experimented with different values for $\beta$ and found the best performing were: $\beta = 1$ for MorphoMNIST, $\beta = 5$ for CelebA and $\beta = 3$ for ADNI. For GAN, we found that fine-tuning with the cyclic cost minimisation detailed in Appendix A.1 was necessary to ensure closeness to the factual image for all datasets.

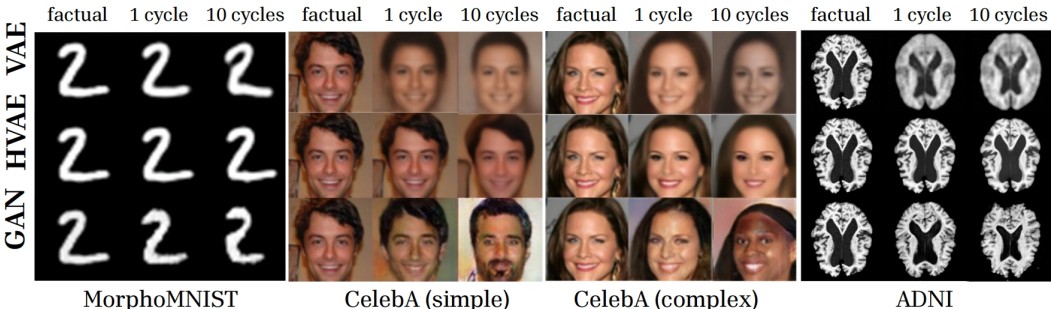

Figure 4: Qualitative evaluation of composition across all datasets/graphs. From left to right across all datasets: (i) factual, (ii) null-intervention (reconstruction) (iii) 10 cycles of null-intervention

## 4.1 Composition

To quantitatively evaluate composition, we perform a *null-intervention*: we abduct the posterior noise and use it to perform inference without intervening on any variable. Following the protocol of (35), we apply composition for 1 and 10 cycles, measuring the distance between the initial observable

image and its first and tenth reconstructed versions, respectively. In addition to the $\ell_1$ distance in the pixel space, we also use the LPIPS (63) metric on VGG-16, since we find that the former is not informative for *higher content* datasets (e.g., natural images here). We empirically confirm that we can effectively capture meaningful differences that align with our qualitative evaluation.

Table 1: Composition metrics for MorphoMNIST, CelebA, and ADNI. For CelebA we include the models trained on the simple as well as on the complex causal graph.

| Model | $l_1$ image space ↓ | | LPIPS ↓ | |
|---|---|---|---|---|
| | 1 cycle | 10 cycles | 1 cycle | 10 cycles |
| **MorphoMNIST** | | | | |
| VAE | $2.600_{1.454}$ | $7.698_{4.199}$ | $0.026_{0.003}$ | $0.075_{0.003}$ |
| HVAE | $\mathbf{0.438_{0.178}}$ | $\mathbf{1.550_{0.541}}$ | $\mathbf{0.006_{0.001}}$ | $\mathbf{0.024_{0.003}}$ |
| GAN | $3.807_{1.822}$ | $11.697_{5.750}$ | $0.049_{0.004}$ | $0.184_{0.008}$ |
| **CelebA (simple/complex)** | | | | |
| VAE | $127.8_{18.1}/\mathbf{121.2_{13.2}}$ | $\mathbf{123.4_{22.1}}/127.9_{21.1}$ | $0.295_{0.004}/0.282_{0.061}$ | $0.424_{0.005}/0.412_{0.091}$ |
| HVAE | $129.4_{11.6}/122.7_{10.4}$ | $138.8_{23.4}/\mathbf{124.6_{16.4}}$ | $\mathbf{0.063_{0.003}}/\mathbf{0.122_{0.033}}$ | $\mathbf{0.200_{0.008}}/\mathbf{0.240_{0.053}}$ |
| GAN | $\mathbf{115.3_{22.0}}/127.6_{17.3}$ | $128.4_{21.4}/131.8_{23.3}$ | $0.290_{0.003}/0.276_{0.074}$ | $0.462_{0.005}/0.490_{0.121}$ |
| **ADNI** | | | | |
| VAE | $18.882_{1.786}$ | $30.250_{3.389}$ | $0.306_{0.008}$ | $0.384_{0.006}$ |
| HVAE | $\mathbf{3.384_{0.367}}$ | $\mathbf{7.456_{0.622}}$ | $\mathbf{0.101_{0.012}}$ | $\mathbf{0.156_{0.014}}$ |
| GAN | $24.261_{1.821}$ | $32.794_{3.578}$ | $0.268_{0.009}$ | $0.323_{0.007}$ |

**MorphoMNIST**: In Table 1 we show quantitative results for composition. We observe that HVAE significantly outperforms all models across most metrics and datasets for both 1 and 10 cycles, followed by VAE. Figure 4 depicts qualitative results for the above. It is evident that the HVAE is capable of preserving extremely well the details of the original image, whereas the GAN introduces the largest distortion after multiple cycles.

**CelebA**: From Table 1 we see that HVAE achieves the best LPIPS distance (on both the simple and complex graph). The efficacy of HVAE is also evident qualitatively in Figure 4: HVAE retains details of the image, whilst VAE progressively blurs and distorts it. GAN does not introduce blurriness, but alters significantly the facial characteristics and the background. Graph size does not seem to play a major role, albeit HVAE shows a small drop in the performance.

**ADNI**: We clearly see that HVAE consistently outperformed the other models both in the $l_1$ and LPIPS distance scores. The results for GAN and VAE are comparable, with GAN performing better in terms of LPIPS. By assessing the qualitative results, it is evident that VAE significantly blurs the original image but keeps its structure, while GAN produces sharp reconstructions which distort important anatomical structures (grey and white matter). HVAE exhibits a similar effect to its VAE counterpart but to a much lesser extent, effectively smoothing the image texture.

**Summary & interpretation**: We find that HVAE consistently outperforms the other models on composition. VAE and GAN were comparable, with VAE reconstructions maintaining structure but being too blurry and GAN changing image structure, especially in complex datasets. The ability of *amortized explicit* methods to retain pixel-level details can be attributed to the reparametrisation they utilise, as we state in Section 3.2 and Appendix A.1. HVAE's advantage to other models becomes evident as data complexity increases (CelebA, ADNI). This might be due to the multiple stochastic layers of latent variables (5), which help to more optimally learn and maintain (across composition cycles) the prior of the latent from the data distribution (7).

## 4.2 Effectiveness

*MorphoMNIST*: To quantitatively evaluate model effectiveness on MorphoMNIST, we train convolutional regressors for continuous variables (thickness, intensity) and a classifier for categorical variables (digits). We perform random interventions on a single parent each time, as per (7). In Table 2, we observe that the mechanism of HVAE achieves the best performance across most counterfactual scenarios. We note that the error for intensity remains low when performing an intervention on thickness, compared to other interventions, which showcases the effectiveness of the conditional intensity mechanism. The compliance with the causal graph of Figure 3 can also be confirmed qualitatively in Figure 5. Intervening on thickness modifies intensity, while intervening on intensity does not affect thickness (graph mutilation). Interventions on the digit alter neither.

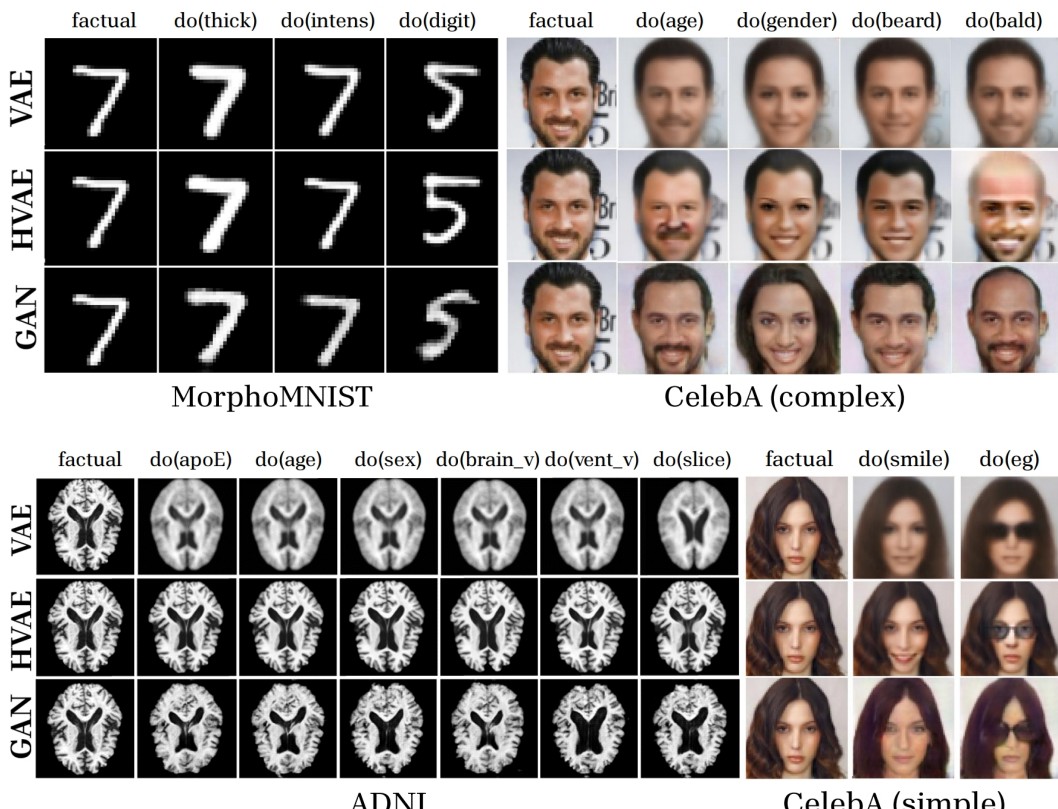

Figure 5: Qualitative evaluation of effectiveness for all datasets/graphs. From left to right across datasets: the leftmost image is the factual one and then each column shows the causal counterfactual image after intervening on a single attribute. v: volume; vent: ventricle; eg: eyeglasses.

Table 2: Effectiveness on MorphoMNIST test set.

| Model | Thickness (t) MAE ↓ | | | Intensity (i) MAE ↓ | | | Digit (y) Acc. ↑ | | |
|---|---|---|---|---|---|---|---|---|---|
| | $do(t)$ | $do(i)$ | $do(y)$ | $do(t)$ | $do(i)$ | $do(y)$ | $do(t)$ | $do(i)$ | $do(y)$ |
| VAE | $0.109_{0.01}$ | $0.333_{0.02}$ | $0.139_{0.01}$ | $3.15_{0.26}$ | $5.33_{0.29}$ | $4.64_{0.35}$ | $\mathbf{0.989_{0.01}}$ | $\mathbf{0.988_{0.01}}$ | $0.775_{0.02}$ |
| HVAE | $\mathbf{0.086_{0.09}}$ | $\mathbf{0.224_{0.03}}$ | $\mathbf{0.117_{0.01}}$ | $\mathbf{1.99_{0.18}}$ | $\mathbf{3.52_{0.26}}$ | $\mathbf{2.10_{0.17}}$ | $0.985_{0.01}$ | $0.935_{0.02}$ | $\mathbf{0.972_{0.02}}$ |
| GAN | $0.228_{0.01}$ | $0.680_{0.02}$ | $0.393_{0.02}$ | $9.43_{0.59}$ | $15.14_{0.99}$ | $12.39_{0.57}$ | $0.961_{0.02}$ | $0.966_{0.01}$ | $0.451_{0.024}$ |

*CelebA*: To measure counterfactual effectiveness for CelebA, we train binary classifiers for each attribute. Following experimentation, we observed that HVAE ignored the conditioning, as reported also in (7). Therefore, we exploited the counterfactual training process proposed in the same paper and summarised in Appendix A.1. Table 3 presents the effectiveness measured in terms of F1 score using each of the attribute classifiers for the simple and complex graph. HVAE outperforms all other models across most scenarios. The most evident difference was observed for *eyeglasses* in the simple graph and for *gender* and *age* in the complex graph. HVAE efficiently manipulated these attributes and generated plausible counterfactuals. Nevertheless, we observe a low F1 score for the *bald* classifier across all possible attribute interventions in the complex graph. This most likely occurs due to the high imbalance of this attribute in the training data, resulting in poor performance.

**ADNI**: Table 4 shows the effectiveness scores for the direct causal parents of the brain image.[4] We used ResNet-18 pretrained on ImageNet (16) to predict these variables given the image. [5] We see that regarding the first two variables, HVAE outperforms VAE and GAN. This can also be qualitatively evaluated on Figure 5, where HVAE correctly lowers the ventricular volume, whereas VAE does not

---

[4]The other three variables exhibit minor differences across models. Their scores can be found in Table 6.

[5]The anti-causal predictor for brain volume is also conditioned on the ventricular volume (Figure 3).

Table 3: Effectiveness on CelebA test set for both graphs.

| | CelebA (simple) | | | |
|---|---|---|---|---|
| **Model** | **Smiling (s) F1** $\uparrow$ | | **Eyeglasses (e) F1** $\uparrow$ | |
| | $do(s)$ | $do(e)$ | $do(s)$ | $do(e)$ |
| VAE | $0.897_{0.02}$ | $0.987_{0.01}$ | $0.938_{0.05}$ | $0.810_{0.02}$ |
| HVAE | $\mathbf{0.998_{0.01}}$ | $\mathbf{0.997_{0.01}}$ | $0.883_{0.06}$ | $\mathbf{0.981_{0.02}}$ |
| GAN | $0.819_{0.02}$ | $0.873_{0.01}$ | $\mathbf{0.957_{0.03}}$ | $0.891_{0.01}$ |

| | CelebA (complex) | | | | | | | |
|---|---|---|---|---|---|---|---|---|
| | **Age (a) F1** $\uparrow$ | | | | **Gender (g) F1** $\uparrow$ | | | |
| | $do(a)$ | $do(g)$ | $do(br)$ | $do(bl)$ | $do(a)$ | $do(g)$ | $do(br)$ | $do(bl)$ |
| VAE | $0.35_{0.04}$ | $0.782_{0.02}$ | $0.816_{0.02}$ | $0.819_{0.02}$ | $0.977_{0.01}$ | $0.909_{0.02}$ | $0.959_{0.02}$ | $\mathbf{0.973_{0.01}}$ |
| HVAE | $\mathbf{0.654_{0.1}}$ | $\mathbf{0.893_{0.04}}$ | $\mathbf{0.908_{0.03}}$ | $\mathbf{0.899_{0.03}}$ | $\mathbf{0.988_{0.02}}$ | $0.949_{0.03}$ | $\mathbf{0.994_{0.01}}$ | $0.95_{0.03}$ |
| GAN | $0.413_{0.04}$ | $0.71_{0.02}$ | $0.818_{0.02}$ | $0.799_{0.01}$ | $0.952_{0.01}$ | $\mathbf{0.982_{0.01}}$ | $0.92_{0.01}$ | $0.961_{0.01}$ |
| | **Beard (br) F1** $\uparrow$ | | | | **Bald (bl) F1** $\uparrow$ | | | |
| | $do(a)$ | $do(g)$ | $do(br)$ | $do(bl)$ | $do(a)$ | $do(g)$ | $do(br)$ | $do(bl)$ |
| VAE | $0.944_{0.01}$ | $0.828_{0.03}$ | $0.296_{0.05}$ | $\mathbf{0.945_{0.02}}$ | $\mathbf{0.023_{0.03}}$ | $0.496_{0.05}$ | $0.045_{0.04}$ | $0.412_{0.03}$ |
| HVAE | $\mathbf{0.952_{0.03}}$ | $\mathbf{0.951_{0.03}}$ | $\mathbf{0.441_{0.11}}$ | $0.916_{0.04}$ | $0.02_{0.05}$ | $\mathbf{0.86_{0.05}}$ | $0.045_{0.07}$ | $\mathbf{0.611_{0.04}}$ |
| GAN | $0.908_{0.01}$ | $0.838_{0.02}$ | $0.233_{0.03}$ | $0.907_{0.01}$ | $0.021_{0.02}$ | $0.82_{0.02}$ | $\mathbf{0.055_{0.02}}$ | $0.492_{0.02}$ |

Table 4: Effectiveness on the ADNI test set. Note that the three direct causal parents to the image are presented: *Brain volume*, *Ventricular volume* and *Slice number*.

| **Model** | **Brain volume (b) MAE** $\downarrow$ | | | **Ventricular volume (v) MAE** $\downarrow$ | | | **Slice (s) F1** $\uparrow$ | | |
|---|---|---|---|---|---|---|---|---|---|
| | $do(b)$ | $do(v)$ | $do(s)$ | $do(b)$ | $do(v)$ | $do(s)$ | $do(b)$ | $do(v)$ | $do(s)$ |
| VAE | $0.17_{0.03}$ | $0.15_{0.06}$ | $0.15_{0.06}$ | $0.08_{0.05}$ | $0.20_{0.04}$ | $0.08_{0.05}$ | $\mathbf{0.52_{0.15}}$ | $\mathbf{0.48_{0.15}}$ | $\mathbf{0.46_{0.10}}$ |
| HVAE | $\mathbf{0.09_{0.03}}$ | $\mathbf{0.12_{0.06}}$ | $\mathbf{0.13_{0.06}}$ | $\mathbf{0.06_{0.04}}$ | $\mathbf{0.04_{0.01}}$ | $\mathbf{0.06_{0.04}}$ | $0.38_{0.15}$ | $0.41_{0.16}$ | $0.41_{0.11}$ |
| GAN | $0.17_{0.02}$ | $0.16_{0.07}$ | $0.16_{0.06}$ | $0.12_{0.02}$ | $0.22_{0.03}$ | $0.12_{0.03}$ | $0.14_{0.03}$ | $0.16_{0.03}$ | $0.05_{0.02}$ |

change it and GAN increases it. In predicting the image slice, VAE performed better with HVAE coming close and GAN having the worst score, especially when intervening on the slice variable.

**Summary & interpretation**: We attribute HVAE's superior performance in generating counterfactuals to the high expressivity of its hierarchical latent variables. This hierarchical structure, allows HVAE to align with the underlying causal graph, thereby retaining semantic information of the original image while effectively implementing the desired causal modifications. The rich representation of latent variables not only preserves original content but also enables more accurate and nuanced manipulation of causal factors, resulting in counterfactuals that are both effective and plausible.

## 4.3 Realism & Minimality (FID & CLD)

The FID metric in Table 10 aligns with our qualitative results: HVAE outperformed both VAE and GAN on MorphoMNIST, CelebA (simple graph) and ADNI. For CelebA (complex graph), GAN produced counterfactuals that were more realistic (while not maintaining identity and background) (Table 10, Figure 5). In terms of minimality (CLD metric) (Table 10), HVAE was the best performing model on CelebA (simple graph) and ADNI, while VAE showed the best outcome on MorphoMNIST and CelebA (complex graph). Appendix A.2 provides further details on the formulation and implementation of both metrics.

**Summary & interpretation**: We found that HVAE generates the most realistic counterfactuals across most metrics and datasets, namely the ones closest to the original data distribution. Introducing more attributes to condition the HVAE (complex graph on CelebA) affected image realism, while GAN maintained good performance. The GAN model, while capable of generating realistic images, failed to achieve optimal minimality due to its lower ability to preserve factual details, compared to HVAE, which is also evident by the composition metric.

Table 5: Realism (FID) and Minimality (CLD) for MorphoMNIST, CelebA, ADNI.

| | **MorphoMNIST** | | **CelebA (simple/complex)** | | **ADNI** | |
|---|---|---|---|---|---|---|
| **Model** | FID $\downarrow$ | CLD $\downarrow$ | FID $\downarrow$ | CLD $\downarrow$ | FID $\downarrow$ | CLD $\downarrow$ |
| VAE | 10.124 | **0.268** | 66.412/59.393 | 0.301/**0.299** | 278.245 | 0.352 |
| HVAE | **5.362** | 0.272 | **22.047**/35.712 | **0.295**/0.305 | **74.696** | **0.347** |
| GAN | 35.568 | 0.286 | 31.560/**27.861** | 0.38/0.304 | 113.749 | 0.353 |

# 5 Discussion and Conclusions

We close a gap in counterfactual image generation, by offering a systematic and unified evaluation framework pitting against diverse causal graphs, model families, datasets, and evaluation metrics. By providing a rigorous and standardized assessment, we can probe the strengths and weaknesses of different approaches, ultimately paving the way for more reliable counterfactual image generation.

In fact, our findings show a superior expressivity of hierarchical latent structures in HVAE, which enabled more accurate abduction of noise variables compared to VAEs and GANs. Such structure can better learn the causal variables of the benchmarked datasets, preserving semantic information and effectively enabling the generation of causal counterfactuals. We identify that the development of Deep-SCM-based conditional HVAE is in the right direction, as it outperformed all other models examined (VAEs, GANs) across most datasets and metrics. Additionally, we extended the evaluation of the above models to previously untested datasets and causal graphs. Specifically, we applied HVAE and GAN to a complex causal graph of human faces, and developed a new GAN architecture, inspired by (59), capable of generating counterfactual brain MRIs conditioned on multiple variables.

**Limitations:** Attempts to adapt the benchmarked models to enable scaling to higher resolution datasets (e.g. CelebAHQ (24)) or to increase the graph complexity for single-variable models (e.g. WGAN of (59), Diff-SCM of (44)) were not fruitful. This is somewhat expected and illustrates the difference between *plain* image editing (less inductive bias) to counterfactual image generation (more inductive bias).

In Figure 8 we can observe that even the best performing model, HVAE, fails to scale to the previously untested resolution of 256x256 (CelebAHQ), preventing a fair comparison. This can be attributed to the counterfactual training step needed to prevent the condition ignoring as described in Appendix A.1. A possible interpretation is that the gradients of the classifiers cannot accurately influence the model output in the pixel space as dimensionality grows. Another challenge for the field resides on the use of model-dependent metrics. For example, it is hard to measure effectiveness accurately. This requires training proxy predictors, which we assume perform as desired. Furthermore, the number of these predictors scales with the size of the causal graph.

**Future work:** We examined three families of generative models conforming to the Deep-SCM paradigm. To perform fair comparisons in the scope of Deep-SCM, we left methods such as deep twin networks (57) and backtracking counterfactuals (29) for future work. Despite their prominence in image generation, diffusion models have not been conditioned yet with non-trivial SCM graphs and don't offer straightforward mechanisms for noise abduction in order to perform counterfactual inference. This means that existing methodologies have to be adapted (44; 43; 12) to accommodate the multi-variable SCM context necessary for representing causal interplay and confounding variables, or new methods must be developed. Having benchmarked existing methods herein, we leave SCM-conditioning for diffusion models, for future work. Nevertheless, we have experimented with the extension of existing methods based on diffusion models for counterfactual image generation. A description of our methodology and preliminary results is included in Appendix E.

We are keen to see how the community will build on our benchmark and expand to novel methods and more complex causal graphs. The user-friendly Python package we provide can be extended to incorporate additional SCMs, causal methods, generative models and datasets to enable such future community-led investigations.

## Acknowledgments and Disclosure of Funding

This work was partly supported by the National Recovery and Resilience Plan Greece 2.0, funded by the European Union under the NextGenerationEU Program (Grant: MIS 5154714). S.A. Tsaftaris acknowledges support from the Royal Academy of Engineering and the Research Chairs and Senior Research Fellowships scheme (grant RCSRF1819/8/25), and the UK's Engineering and Physical Sciences Research Council (EPSRC) support via grant EP/X017680/1, and the UKRI AI programme and EPSRC, for CHAI - EPSRC AI Hub for Causality in Healthcare AI with Real Data [grant number EP/Y028856/1]. P. Sanchez thanks additional financial support from the School of Engineering, the University of Edinburgh.

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

# A  Theory

## A.1  Mechanisms and Models

We consider three categories of invertible mechanisms:

1. **Invertible, explicit** used for the attribute mechanisms in conjunction with *Conditional Normalising Flows* (53). This mechanism is invertible by design.

2. **Amortised, explicit** employed for high-dimensional variables, such as images. This can be developed with Conditional VAEs (25; 20) or extensions like Conditional Hierarchical Variational Autoencoder (HVAE) (26; 47). The structural assignment is decomposed into a low-level invertible component, in practice the reparametrisation trick, and a high-level non-invertible component that is trained in an amortized variational inference manner, which in practice can be a probabilistic convolutional decoder. This also entails a noise decomposition of $\epsilon_i$ into two noise variables $(\epsilon, z)$, s.t. $P(\epsilon_i) = P(\epsilon)P(z)$.

3. **Amortised, implicit** This mechanism can also be employed for the imaging variables. However, it does not rely on approximate maximum-likelihood estimation for training, but optimises an adversarial objective with a conditional implicit-likelihood model. It was introduced in (6) and developed further with Conditional GANs (33; 10).

The models we use to implement these mechanisms are detailed below.

**Conditional Normalising Flows** To enable tractable abduction of $pa_x$, invertible mechanisms can be learned using conditional normalising flows. A normalising flow maps between probability densities defined by a differentiable, monotonic, bijection. These mappings are a series of relatively simple invertible transformations and can be used to model complex distributions (51).

In particular, a random variable $z_0$ is mapped through a normalising flow $f$, to a transformed variable $z_K = f_K(f_{K-1}(\ldots f_1(z_0)))$ and its distribution $p(z_K) = p(z_0) - \sum_{k=1}^{K} |\det df_k(z_{k-1})| \, dz_{k-1}$, where $p(z_0)$ is a standard normal base distribution and each $f_k$ is a simple, monotonically increasing function which has a closed form with an easy to calculate derivative.

The mechanism for each attribute $(pa_x)$ is a normalising flow $f(\epsilon)$. The base distribution for the exogenous noise is typically assumed to be Gaussian $\epsilon \sim N(0, I)$. The distribution of $pa_x$ can be computed as $P(pa_x) = P(\epsilon) |\det \nabla f(\epsilon)|^{-1}$. Thus, the conditional distribution $P(pa_x | pa_{pa_x})$ can be computed as:

$$P(pa_x | pa_{pa_x}) = p(\epsilon) |\det \nabla f(\epsilon; pa_{pa_x})|^{-1} \tag{1}$$

**Conditional VAE** Conditional VAEs (25; 20) are composed of a generative model $p_\theta(x, z | pa_x) = p_\theta(x | z, pa_x)p(z)$ and an inference model $q_\phi(z | x, pa_x)$, where $pa_x$ is the conditioning signal. The likelihood $p_\theta(x | z, pa_x)$ and the approximate posterior $q_\phi(z | x, pa_x)$ are assumed to be diagonal Gaussians, while the prior $p(z)$ is fixed as standard Gaussian $p(z) \sim N(0, I)$. The evidence lower bound loss can be expressed as:

$$\text{ELBO}_\beta(\phi, \theta) = \mathbb{E}_{z \sim q_\phi(z | x, pa_x)}[p_\theta(x | z, pa_x)] - \beta D_{KL}[q_\phi(z | x, pa_x) || p(z)] \tag{2}$$

The distributions $q_\phi(z | x, pa_x)$, $p_\theta(x | z, pa_x)$ are parameterised as neural networks with parameters $\phi$ and $\theta$ accordingly that output the $\mu$ and $\sigma$ of the distributions. The ELBO is jointly maximised with respect to $\phi, \theta$.

To produce counterfactuals via leveraging conditional VAEs (25; 20), the high-dimensional variable $x$ along with its factual parents $pa_x$ are encoded into a latent representation $z \sim q_\phi(z | x, pa_x)$ which is sampled through the reparameterisation trick: $z = \mu_\phi(x, pa_x) + \sigma_\phi(x, pa_x) * \epsilon$, where $\epsilon \sim N(0, I)$ (7). Then, the latent variable $z$ and the counterfactual parents $pa_x^*$ are decoded to obtain the counterfactual distribution $(\mu_\theta^*, \sigma_\theta^*)$. Since the observational distribution is assumed to be Gaussian, sampling can be performed by forwarding the decoder with the factual parents and then applying a reparametrization: $x = \mu_\theta(z, pa_x) + \sigma_\theta(z, pa_x) * \epsilon, \epsilon \sim N(0, I)$. Thus, the exogenous noise can be expressed as $\epsilon = \frac{x - \mu_\theta(z, pa_x)}{\sigma_\theta(z, pa_x)}$. Finally, this $\epsilon$ is used to sample from the counterfactual distribution as follows: $x^* = \mu_\theta(z, pa_x^*) + \sigma_\theta(z, pa_x^*) * \epsilon$.

**Conditional HVAE** Hierarchical VAEs (26; 47) extend classical VAEs by introducing $L$ groups of latent variables $\boldsymbol{z} = \{z_1, z_2, ..., z_L\}$. One widely used variation of the HVAE is the top-down VAE

which is proposed in (47). In the top-down approach (47; 5; 55) both the prior and the approximate posterior emit the groups of the latent variables in the same top-down order, thus the two distributions can be factorised as:

$$p_\theta(z_{1:L}) = p_\theta(z_L)p_\theta(z_{L-1}|z_L)...p_\theta(z_1|z_{i>1}), \tag{3}$$

$$q_\phi(z_{1:L}|x) = q_\phi(z_L|x)q_\phi(z_{L-1}|z_L,x)...q_\phi(z_1|z_{i>1},x) \tag{4}$$

A conditional structure of the top-down HVAE (47; 5; 55) can be adopted to improve the quality of produced samples and abduct noise more accurately. As we consider Markovian SCMs, the exogenous noise $\epsilon$ has to be independent, therefore as the latents $z_{1:L}$ are downstream components of the exogenous noise, the corresponding prior $p_\theta(z_{1:L})$ has to be independent from $pa_x$. In order to accomplish this, the authors of (7) propose to incorporate $z_i$ and $pa_x$ at each top-down layer through a function $f_\omega^i$ which is modelled as a learnable projection network. Particularly, the output of each top-down layer can be written as:

$$h_i = h_{i+1} + f_\omega^i(z_i, pa_x), \quad z_i \sim p_\theta(z_i|z_{>i}), \quad i = L-1, ..., 1, \tag{5}$$

where $h_{init}$ can be learned from data. Thus, the joint distribution of $x, z_{1:L}$ conditioned on $pa_x$ can be expressed as:

$$p_\theta(x, z_{1:L}|pa_x) = p_\theta(x|z_{1:L}, pa_x)p_\theta(z_{1:L}), \tag{6}$$

while the conditional approximate posterior is defined as:

$$q_\phi(z_{1:L}|x, pa_x) = q_\phi(z_L|x, pa_x)q_\phi(z_{L-1}|z_L, x, pa_x)...q_\phi(z_1|z_{i>1}, x, pa_x) \tag{7}$$

Following a methodology analogous to standard VAE, the exogenous noise can be abducted as $\epsilon = \frac{x - \mu_\theta(z_{1:L}, pa_x)}{\sigma_\theta(z_{1:L}, pa_x)}$. Therefore, a counterfactual sample can be obtained as: $x^* = \mu_\theta(z_{1:L}, pa_x^*) + \sigma_\theta(z_{1:L}, pa_x^*) * \epsilon$.

**Counterfactual training** This fine-tuning step was proposed in (7) to alleviate the ignoring of the conditioning that happened for some datasets. To achieve this, the model generates counterfactuals and utilises the losses of classifiers trained on observational data, while keeping the reconstruction loss above a threshold. In brief, the lowest ELBO loss is incorporated early during training into the counterfactual loss by introducing a Lagrange coefficient, thus transforming the setup into a Lagrangian optimisation problem (41). The overall objective can be expressed as (7):

$$L(\theta, \phi, \lambda; \mathbf{x}, \mathbf{pa_x}) = -\sum_{k=1}^{K} \mathbb{E}_{\substack{\mathbf{pa_{x_k}^*} \sim p(\mathbf{pa_{x_k}}) \\ \mathbf{x}^* \sim P_{\mathbf{CF}}(\mathbf{x}^*|do(\mathbf{pa_{x_k}^*}), \mathbf{x})}} \left[ \log q_{\psi_k}(\mathbf{pa_{x_k}^*} \mid \mathbf{x}^*) \right]$$
$$-\lambda(c - ELBO(\theta, \phi; \mathbf{x}, \mathbf{pa_x})), \tag{8}$$

where $q_{\psi_k}$ is the $k_{th}$ parent classifier, $c$ is the ELBO that the model achieves during pretraining and $P_{CF}$ the counterfactual distribution. To optimize the above objective gradient descent is performed on the HVAE parameters $\phi, \theta$ and gradient ascent on the Lagrange coefficient $\lambda$.

It should be noted that this process can introduce bias from the classifiers, which bias is in any case present in the data and can be picked up from the model even without this step. Finally, using the same classifiers to measure effectiveness is unfair compared to the other models, which we alleviate by using different classifiers to guide the counterfactual finetuning.

**Conditional GAN** Finally, we also consider the conditional GAN-based framework (6) for counterfactual inference. This setup includes an encoder $E$, that given an image $x$ and its parent variables $pa_x$ produces a latent code $z_x = E(x, pa_x)$. The generator $G$ then receives either a noise latent $z$ or the produced $z_x$ together with the parents $pa_x$ to produce an image $x' = G(z', pa_x)$. The image alongside the latent and the parents $pa_x$ are then fed into the discriminator. The encoder and the generator are trained together as in (10) to deceive the discriminator in classifying generated samples from a noise latent $z$ as real, while classifying samples produced from $z_x$ (real image latents) as fake, while the discriminator has the opposite objective. Providing $pa_x$ as input to all models is proven to make conditioning on parents stronger. The conditional GAN is optimised as follows:

$$\min_{E,G} \max_D V(D, G, E) = \mathbb{E}_{q(x)p(pa_x)}[log(D(x, E(x, pa_x), pa_x))]$$
$$+ \mathbb{E}_{p(z)p(pa_x)}[log(1 - D(G(z, pa_x), z, pa_x))], \tag{9}$$

where $q(x)$ is the distribution for the images, $p(z) \sim N(0, I)$ and $p(pa_x)$ is the distribution of the parents. To produce counterfactuals images, the factual $x$ and its parents $pa_x$ are given to the encoder which produces the latent $z_x$ which serves as the noise variable. Then, the counterfactual parents $pa_x^*$ and $z_x$ are fed into the generator to produce the counterfactual image $x^*$. This procedure can be formulated as:

$$x^* = G(E(x, (pa_x)), pa_x^*)), \tag{10}$$

However, as it is discussed in the original paper (6) and as we confirmed empirically, this objective is not enough to enforce accurate abduction of the noise, resulting in low reconstruction quality. In order to alleviate this, the encoder is fine-tuned, while keeping the rest of the network fixed, to minimise explicitly the reconstruction loss on image and latent space. This cyclic cost minimisation approach, introduced by (9), enables accurate learning of the inverse function of the generator.

$$\text{Error in the image space } L_x = \mathbb{E}_{x \sim q(x)} \|x - G(E(x, pa_x), pa_x)\|_2$$
$$\text{Error in the latent space } L_z = \mathbb{E}_{z \sim p(z)} \|z - E(G(z, pa_x), pa_x)\| \tag{11}$$

## A.2  Metrics

**Composition** *If we force a variable $X$ to a value $x$ it would have without the intervention, it should have no effect on the other variables* (13). It follows then, that we can have a *null-intervention* that should leave all variables unchanged. Based on this property one can apply the *null-intervention $m$* times, denoted as $f_\emptyset^m$, and measure the $l_1$ distance $d(\cdot)$ between the $m$ produced and the original image (35). To produce a counterfactual under the *null-intervention*, we abduct the exogenous noise and skip the action step.

$$\text{composition}^m(x, pa_x) = d(x, f_\emptyset^m(x, pa_x)) \tag{12}$$

As the $\ell_1$ distance on image space yields results that do not align with our perception and favours shortcut features such as simpler backgrounds, we extend composition by computing the Learned Perceptual Similarity (LPIPS) (63).

**Effectiveness** aims to identify how successful is the performed intervention. In other words, *if we force a variable $X$ to have the value $x$, then $X$ will take on the value $x$* (13). In order to quantitatively evaluate effectiveness for a given counterfactual image we leverage an anti-causal predictor $g_\theta^i$ trained on the data distribution, for each parent variable $pa_x^i$ (35). Each predictor, then, approximates the counterfactual parent $pa_x^{i*}$ given the counterfactual image $x^*$ as input

$$\text{effectiveness}_i(x^*, pa_x^{i*}) = d(pa_x^{i*}, g_\theta^i(x^*)), \tag{13}$$

where $d(\cdot)$ is a distance, defined as a classification metric for categorical variables and as a regression metric for continuous ones.

**Realism** For realism we use the *Fréchet Inception Distance* (FID) (19) which captures the similarity of the set of counterfactual images to all images in the dataset. Real and counterfactual samples are fed into an Inception v3 model (50) trained on ImageNet, in order to extract their feature representations, that capture high-level semantic information.

We denote as $q$ the distribution of the feature representations of counterfactuals and as $p$ the distribution of the feature representations of the factuals. Fréchet distance $d$ is defined as the distance between the Gaussian with mean $(m_q, C_q)$ obtained from $q$ and the Gaussian with mean $(m_p, C_p)$ obtained from $p$. FID is defined as:

$$d^2((m_q, C_q), (m_p, C_p)) = \|m_q - m_p\|_2^2 + \text{Tr}(C_q + C_p - 2(C_q C_p)^{\frac{1}{2}}). \tag{14}$$

**Minimality** The need for minimality of counterfactuals can be justified on the basis of the sparse mechanism shift hypothesis (46). While we can achieve an approximate estimate of minimality by combining the composition and effectiveness metrics, we find that an additional metric is helpful to measure the closeness of the counterfactual image to the factual. For this reason, we leverage a modified version of the *Counterfactual Latent Divergence* (CLD) metric introduced in (44):

$$\text{CLD} = \log(w_1 \exp P(S_{x^*} \leq div) + w_2 \exp P(S_x \geq div)), \tag{15}$$

where $div$ is the distance of the counterfactuals $x^*$ from the factuals $x$, $S_x$, $S_{x^*}$ are the sets of all distances of the factual image to the images that have the same label as the factual and the counterfactual, respectively: $S_x = \{d(x, x')|pa_{x'} = pa_x\}$, $S_{x^*} = \{d(x, x')|pa_{x'} = pa_{x^*}\}$, where $x'$ is any image from the data distribution and $pa_{x'}$ is the variable being intervened upon. To compute the distance $d(\cdot)$, we train an unconditional VAE and use the KL divergence between the latent distributions[6]: $d(x_i, x_j) = D_{KL}(\mathcal{N}(\mu_i, \sigma_i), \mathcal{N}(\mu_j, \sigma_j))$. This metric is minimised by keeping both probabilities low, which represent a trade-off between being far from the factual class but not as far as other (real) images from the counterfactual class are. The weights $w_1$, $w_2$ depend on the effective difference each variable has on the image.

# B    Experimental Setup

## B.1    Dataset preprocessing and splits

The train set of MorphMNIST consists of 48000 samples, while validation and test set consist of 12000 and 10000 samples, respectively. For the CelebA dataset, we used 162770 images for the training, while we kept 19867 images for the validation set and 19962 images for the counterfactual test set. For the ADNI dataset, after pre-processing (following the process of (59)) we get 2D axial grayscale slice images of 192x192 resolution. For each subject we use all visits (2-4 visits depending on the patient) and take the 20 middle MRI slices, ending up with a train set of 10780 images (180 subjects in total) and a test set of 2240 images (38 subjects).

## B.2    Normalising Flows

For Normalising Flows we leverage the normflows package (51) and base our implementation in (27). We use the Adam optimizer with learning rate $10^{-3}$, batch size of 64 and early stopping with the patience parameter set to 10.

## B.3    VAE

We adopt the VAE implementation as defined in (36; 7). The output of the VAE decoder is passed though a Gaussian network that is implemented as a convolutional layer to obtain the corresponding $\mu$, $\sigma$. Then, the reconstructed image is sampled with the reparameterisation trick. To condition the model we concatenate the parent variables with the output features of the first Linear layer of the encoder. We train the model until convergence with a batch size of 128 leveraging the Adam optimizer with initial learning rate $5 \cdot 10^{-4}$ and $\beta_1$=0.9, $\beta_2$=0.999. We also use a linear warmup scheduler for the first 100 iterations. Additionally, we set the early stopping patience to 10 and add gradient clipping and gradient skipping with values 350 and 500 respectively. The dimension of the latent variable is set to 16.

## B.4    HVAE

We follow the experimental setup of (7) leveraging the proposed conditional structure of the very deep VAE (5). To condition the model we expand the parents and concatenate them with the corresponding latent variable at each top-down layer.

For MorphoMNIST, we use a group of 20 latent variables with spatial resolutions $\{1, 4, 8, 16, 32\}$. Each resolution includes 4 stochastic blocks. The channels of the feature maps per resolution are $\{256, 128, 64, 32, 16\}$, while the dimension of the latent variable channels is 16.

We train the model until convergence (approximately 1M iterations) with a batch size of 256 and apply early stopping with patience 10. The resolution of the input images is 32x32. We use the AdamW optimizer with initial learning rate $10^{-3}$, $\beta_1 = 0.9$, $\beta_2 = 0.9$, a weight decay of 0.01 and a linear warmup scheduler for the first 100 iterations. We additionally add gradient clipping and gradient skipping with values 350 and 500, respectively.

For the CelebA experiments we slightly extend the architecture in order to be functional with 64x64 images. Particularly, the spatial resolutions are set to $\{1, 4, 8, 16, 32, 64\}$, while the widths per

---

[6]We computed CLD on the embedding space of a separately trained unconditional VAE for each dataset. We found other distances (LPIPS, CLIP) to perform similarly.

resolution are $\{1024, 512, 256, 128, 64, 32\}$. We consider 4 stochastic blocks per resolution leading to a total of 24 latent variables. Finally, we also set the channels of the latent variables to 16.

As we have discussed in the main paper we employ a two phase training which consists of a pretraining stage and a counterfactual fine-tuning stage. We pretrain the model for 2M iterations with a batch size of 32 using the AdamW optimizer with initial learning rate $10^{-3}$, $\beta_1$=0.9, $\beta_2$=0.9, weight decay 0.01, and early stopping patience set to 10. We use a linear warmup scheduler for the first 100 iterations and also set values for gradient clipping and gradient skipping to 350 and 500 accordingly. We further fine-tune the model for 2K iterations with a initial learning rate of $10^{-4}$ and the same batch size and warmup scheduler, while we use a distinct AdamW optimizer with learning rate 0.01 to perform gradient ascent on the Lagrange coefficient.

For ADNI, we utilised the architecture used in (7) for another dataset of brain MRIs (UK Biobank (49)). We train the model with the same hyperparameters detailed above for CelebA. We find that the counterfactual fine-tuning step is not needed since the model can produce counterfactuals and does not learn to just reproduce the original image. We tried adding this step, but found it to harm the performance of the model.

## B.5   GAN

For MorphoMNIST and CelebA experiments, we adopt the experimental setup described in (6). To condition the model, we directly pass the parent variables into the networks by concatenating them with the image for the encoder and discriminator, and with the latent variable for the generator. For both training and fine-tuning, we use the Adam optimizer with initial learning rate $10^{-4}$, $\beta_1$=0.5, $\beta_2$=0.999 and batch size of 128. We implement early stopping with a patience of 10 epochs.

In order to monitor the generative abilities of the model for the MorphoMNIST dataset, during training, we compare the factual images $x$ with the generated samples produced from a random noise latent $z$ and the ones produced from the latent representations of the encoder. We do this by measuring the $l_1$ distance between their embeddings, which we get by concatenating the last layer activations across all our predictors. For monitoring the fine-tuning of the GAN encoder, we measure the $l_1$ distance between the factual image embeddings and the ones of the samples generated by the encoder-produced latents.

For CelebA, we monitor the generative abilities of the model during training by measuring the FID score of the samples generated from random noise latent variables $z$ and latents produced by the encoder. As in the experiments on MorphoMNIST, we monitor fine-tuning in CelebA using the LPIPS metric (on VGG-16) between real images and samples generated by the encoder-produced latents.

For ADNI, we initially tried adding more and wider layers to the architecture used for CelebA but the model would not converge even with extensive tuning of the hyperparameters. We modified the architecture of (59), which was used to generate brain MRIs, intervening on the age in the ADNI dataset. We should note that the original architecture and training procedure would learn to reproduce the image and therefore did not work with a complex SCM as in our case. We removed the residual connections of the generator and modified the conditioning mechanism to use a concatenation of three variables (encoded with Fourier embeddings as in (3)). Finally, we used the losses we discuss for Conditional GAN above to make for a fairer comparison across datasets.

The rest training parameters for CelebA and ADNI experiments adhere to the same configuration as described for MorphoMNIST.

## B.6   Anti-causal predictors

For the MorphoMNIST attributes, we train deep convolutional regressors for continuous variables and a deep convolutional classifier for the *digit*, using the implementation of (7). For CelebA (simple/complex) and ADNI attributes we used ResNet-18 pretrained on ImageNet. Additionally, as CelebA is a highly unbalanced dataset, similarly to (35) we choose to use a weighted sampler to ensure a balanced class distribution in each training batch. Moreover, the anti-causal predictor of parent variables, are conditioned on all their children variables, including the image. All predictors were trained with batch size 256 using the AdamW optimizer with initial learning rate $10^{-3}$ and a linear warmup scheduler for the first 100 iterations.

# C  Additional qualitative results

## C.1  Composition

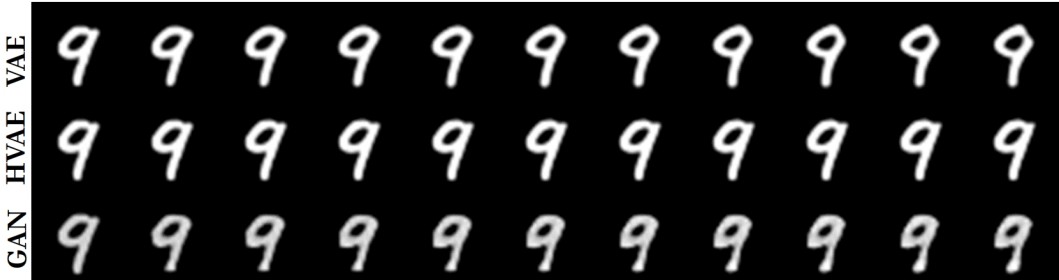

MorphoMNIST

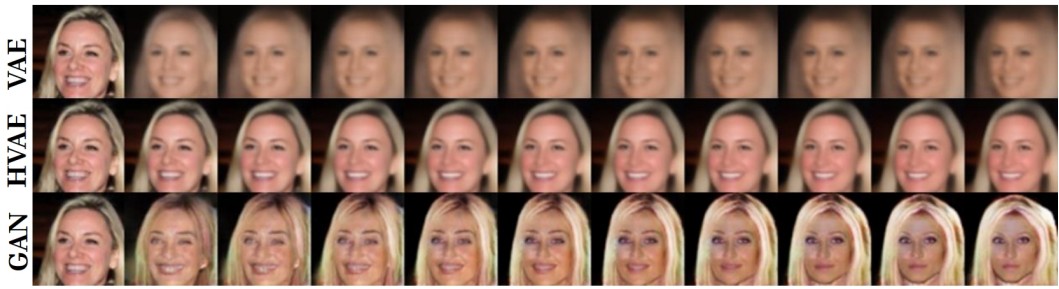

CelebA (simple)

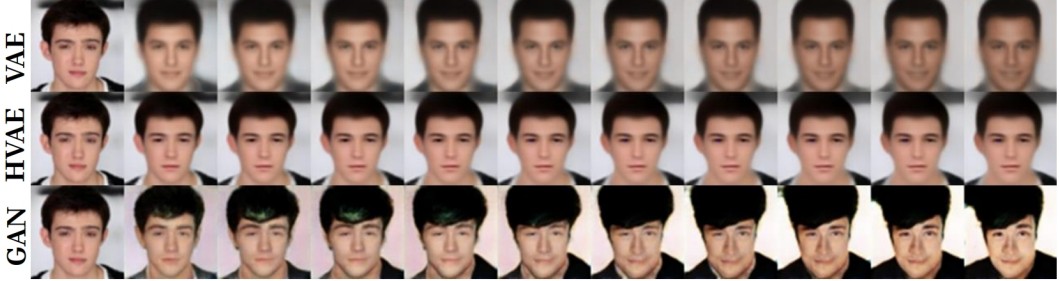

CelebA (complex)

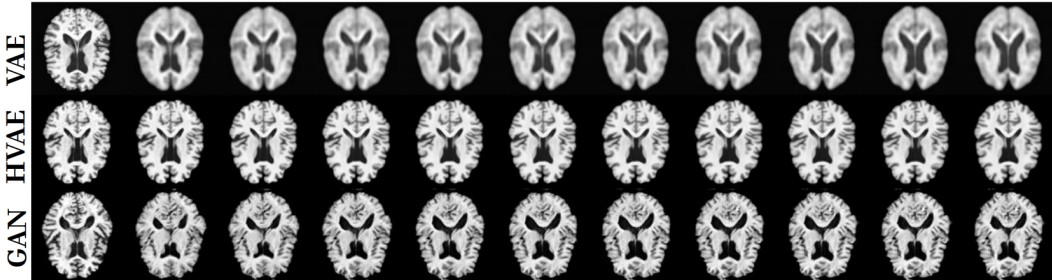

ADNI

Figure 6: Composition for all datasets (10 cycles)

## C.2 Effectiveness

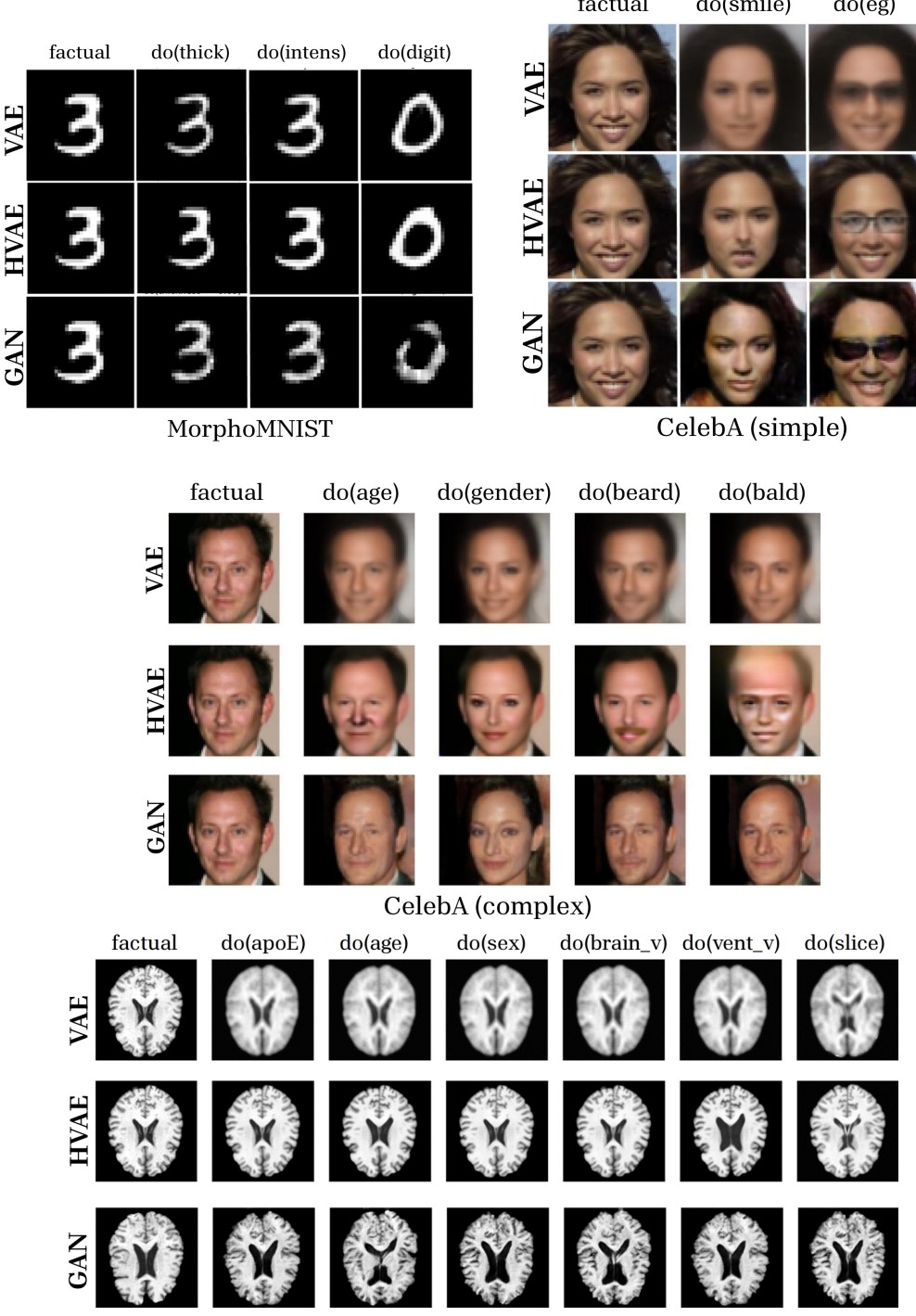

Figure 7: Effectiveness qualitative results for all datasets

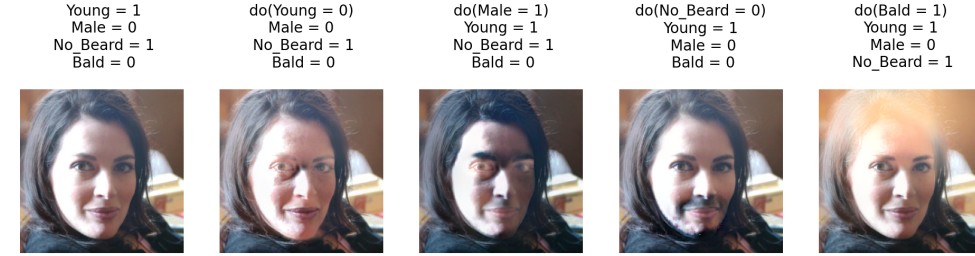

Figure 8: Celeba-HQ (complex) (resolution 256x256) with HVAE. Counterfactual training produces unrealistic artifacts on higher-resolutions.

# D    Additional quantitative results

Table 6: Effectiveness on ADNI test set for all variables.

| Model | F1 $\uparrow$ $do(apoE)$ | MSE $\downarrow$ $do(age)$ | F1 $\uparrow$ $do(sex)$ | MSE $\downarrow$ $do(brain_{vol})$ | MSE $\downarrow$ $do(vent_{vol})$ | F1 $\uparrow$ $do(slice)$ |
|---|---|---|---|---|---|---|
| VAE | $0.278_{0.19}$ | $0.169_{0.04}$ | $0.589_{0.35}$ | $0.167_{0.03}$ | $0.199_{0.04}$ | $0.408_{0.10}$ |
| HVAE | $0.280_{0.19}$ | $0.170_{0.03}$ | $0.589_{0.35}$ | $\mathbf{0.096_{0.03}}$ | $\mathbf{0.037_{0.01}}$ | $\mathbf{0.409_{0.13}}$ |
| GAN | $0.291_{0.237}$ | $0.151_{0.037}$ | $0.544_{0.365}$ | $0.173_{0.04}$ | $0.223_{0.036}$ | $0.045_{0.044}$ |

Table 7: F1 Scores of classifers on CelebA validation set.

| Attribute (F1) $\uparrow$ | ResNet-18 | Standard CNN |
|---|---|---|
| Age | 0.921 | 0.901 |
| Gender | 0.987 | 0.943 |
| Beard | 0.972 | 0.932 |
| Bald | 0.721 | 0.642 |

# E    Diffusion models

Here we describe our efforts to extend an existing method that performs counterfactual image generation using diffusion models, but does so for causal graphs with a single variable. We adapted the classifier-free guidance setting of (43) in order to model the causal graphs of the datasets we are benchmarking. Initially this method under-performed (which we confirmed with the authors as a limitation of their model) when modeling non-trivial graphs (with multiple variables), which can be partially attributed to the spurious correlation of the variables. Also, we empirically observed that the nature of the conditioning mechanisms used and the architectural and hyperparameter choices utilised, can make an important difference for the performance of diffusion models.

We incorporated a UNet2DConditionModel taken from the diffusers library (40) in our codebase and tested it using our framework. We followed the same setting as in all our models, treating the forward diffusion process as the noise abduction and the backward process as the prediction.

We include experiments on MorphoMNIST, as well as CelebA (simple). We report that for MorphoMNIST, the Diffusion model performs worse than the rest of the models in terms of composition and realism (FID), while its performance on effectiveness and minimality (CLD) is comparable to that of the GAN. This is somehow expected, as this dataset is not complex in terms of image content and our simpler models, such as VAE perform well. For CelebA (using the simple causal graph), the diffusion model performs excellent on composition for 1 cycle, but it deteriorates significantly after more cycles. Regarding effectiveness, its performance is slightly worse than the other models, but ranks second when measuring its effect of intervening on eyeglasses. Finally, the realism and minimality of generated images is fairly good but worse than the HVAE.

We believe that the underperformance of the current Diffusion model in the task we are benchmarking is related to architectural and hyperparameter choices utilised. One of them can be the cross-attention

conditioning mechanism we incorporated (between the attributes and the image). As this extension is a naive approach, we believe that novel methodological contribution is needed to enable a more fair comparison with other generative models that have been already proposed for counterfactual image generation in this setting.

Table 8: Composition of Diffusion model with cross-attention on MorphoMNIST and CelebA (simple).

| Model | MorphoMNIST ($L_1$ image space) | | CelebA (LPIPS) | |
|---|---|---|---|---|
| | 1 cycle | 10 cycles | 1 cycle | 10 cycles |
| Diffusion | $17.978_{3.310}$ | $18.963_{3.931}$ | $0.081_{0.010}$ | $0.542_{0.014}$ |

Table 9: Effectiveness of Diffusion model with cross-attention on MorphoMNIST and CelebA (simple).

| MorphoMNIST | | | | | | | | | |
|---|---|---|---|---|---|---|---|---|---|
| Model | Thickness (t) MAE ↓ | | | Intensity (i) MAE ↓ | | | Digit (y) Acc. ↑ | | |
| | do(t) | do(i) | do(y) | do(t) | do(i) | do(y) | do(t) | do(i) | do(y) |
| Diffusion | $0.183_{0.005}$ | $0.281_{0.01}$ | $0.163_{0.005}$ | $12.598_{1.033}$ | $19.267_{1.009}$ | $13.859_{0.651}$ | $0.848_{0.015}$ | $0.791_{0.015}$ | $0.563_{0.01}$ |
| CelebA (simple) | | | | | | | | | |
| | Smiling (s) F1 ↑ | | | | Eyeglasses (e) F1 ↑ | | | | |
| | $do(s)$ | | $do(e)$ | | $do(s)$ | | $do(e)$ | | |
| Diffusion | $0.823_{0.086}$ | | $0.738_{0.153}$ | | $0.863_{0.008}$ | | $0.910_{0.036}$ | | |

Table 10: Realism (FID) and Minimality (CLD) of Diffusion model with cross-attention on MorphoMNIST and CelebA (simple).

| Model | MorphoMNIST | | CelebA (simple) | |
|---|---|---|---|---|
| | FID ↓ | CLD ↓ | FID ↓ | CLD ↓ |
| Diffusion | 60.349 | 0.282 | 29.620 | 0.299 |

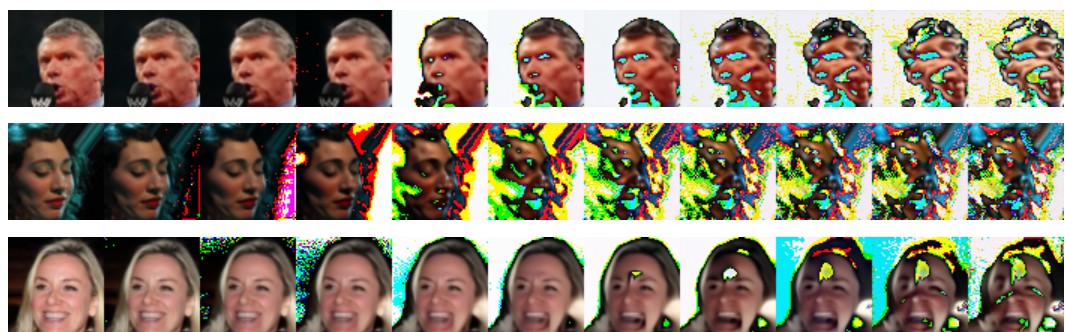

Figure 9: Composition (10 cycles) for CelebA (simple) with Diffusion Model (conditioning with cross-attention)

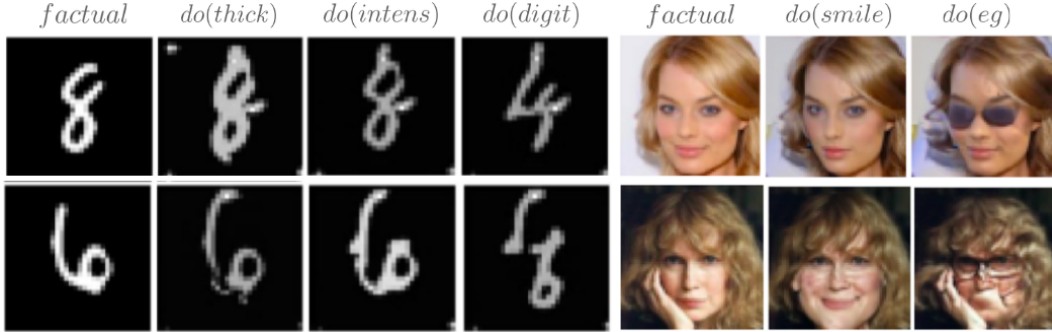

Figure 10: Effectiveness for MorphoMNIST and CelebA (simple) with Diffusion Model (conditioning with cross-attention)

