# OpenReview forum: "Benchmarking Counterfactual Image Generation"
_NeurIPS.cc/2024/Datasets_and_Benchmarks_Track — NeurIPS 2024 Track Datasets and Benchmarks Poster_

### Official Review · Reviewer_F7FH · 2024-07-19
**A valuable benchmark to evaluate counterfactual image generation methods**

**Rating:** 8
**Confidence:** 4
**Correctness:** I my view all claims are correct.
**Clarity:** The paper is well written and easy to…

**Review:**

I believe that this is the first benchmark of its kind and thus particularly valuable. I like the work a lot and this it is a great contribution to the community.
Pro:
- New benchmark for an up and coming field
- Source code looks reasonable clean and easy to work with

Cons
- I would have prefered a more in-depth and detailed discussion of the metrics. Why are they good metrics? I think putting their definition into the appendix is not ideal. I would rather move some parts of the results to the appendix. Fundamentally the metrics and their justification are much more important than some results that will be outdated in a few month anyway.

**Strengths:**

I think the benchmark and the source code are well crafted, timely and relevant and thus deserve publication.

**Additional Feedback:**

-

**Documentation:**

Documentation looks good and the code is reasonably clean.

**Ethics:**

I see no ethical problems.

**Limitations:**

I think the limitations are sufficiently stated.

**Opportunities For Improvement:**

My main point of improvement would be to focus the text much more on the metrics and especially justify why the chosen metrics make sense. What does, for example, "realism" mean, why is comparing it to other, potentially unrelated, imaged a good way of measuring it? Especially the problem of realism and internal consistency is a problem for AI-generated images, so it would deserve more space in the paper.
To make space, I'd move most or even all of the tables into the appendix.

**Relation To Prior Work:**

I think the authors do a superb job of engaging with related work and provide an extensive bibliography.

**Summary And Contributions:**

This paper introduces a benchmark to compare counterfactual image generation methods, i.e. methods that rely on a structural causal model to guide image manipulation.  To the best of my knowledge this is the first benchmark of its kind and thus especially interesting.

---

> ### Author Rebuttal · Authors · 2024-08-17
>
> *I would have preferred a more in-depth and detailed discussion of the metrics. Why are they good metrics? I think putting their definition into the appendix is not ideal. I would rather move some parts of the results to the appendix. Fundamentally the metrics and their justification are much more important than some results that will be outdated in a few month anyway.*
>
> We agree with the reviewer that it is better for the metrics to be a part of the main paper, therefore we will augment Section 3.3 with the detailed descriptions of Appendix A.2 in the camera-ready.
> The same applies to the process of counterfactual training of the HVAE, for which we will move key details from Appendix A.1 to the main paper. We will also add more details on the second paragraph of Section 3.2 that describes how counterfactuals are obtained for all models.
> We will use the extra 1 page (to be provided), as well as extra space by merging tables 2-4 into one.
>
> *What does, for example, "realism" mean, why is comparing it to other, potentially unrelated, imaged a good way of measuring it?*
>
> Realism is the most widely used and sought after metric when it comes to image generation.
> It has been introduced in \cite{fid} to capture the similarity of generated images to real ones (images from the train dataset) as an improvement to the previously used Inception score.
> As we explain in the paper, to measure this score we compare the feature distribution of the generated images with all images belonging to the original dataset.
> It is interesting though when we want to measure the realism of  counterfactuals. Sometimes they may not be realistic, for example when intervening with *do(Beard=True)* on a female. The resulting image is not realistic in the sense that the women in the dataset do not have beards. However, FID does measure the realism of the entire image (e.g. background, eyes) and since we perform the same interventions for all models, the comparison is fair.

---

> ### Author Response · Authors · 2024-08-23
> **Follow up**
>
> Dear Reviewer F7FH,
>
> Thank you again for you hard work in reviewing this paper. We just wanted to follow up to see if you've had a chance to review and consider our response. Do let us know if it addresses your concerns, and please let us know if you have further suggestions or remaining concerns.

---

> > ### Comment · Reviewer_F7FH · 2024-08-29
> >
> > Thank you and sorry for the belated reply - holiday season:)
> >
> > Your comments address my concern, thank you!

---

> > > ### Author Response · Authors · 2024-08-29
> > >
> > > We would like to thank again the reviewer for their hard work in reviewing our work and the positive comments that we addressed all their concerns and feedback.

---

### Official Review · Reviewer_HeXk · 2024-07-22
**Review of Benchmarking Counterfactual Image Generation**

**Rating:** 7
**Confidence:** 4
**Correctness:** Yes.
**Clarity:** Yes.

**Review:**

The quality, clarity, and significance of this work are good. The pros of the work include soundness supported by sufficient experimental results and the interesting conclusions of the study. While the cons are: both the low resolution of the images and the limited range of generated categories would prejudice the study results. And the proposed method has limitations and fails to analyze some currently representative editing methods like denoising diffusion models.

**Strengths:**

1. The authors conduct abundant experiments on various datasets with several methods, supporting the work's soundness.
2. The benchmarking study brings useful and interesting conclusions: HVAE performs better in composition, effectiveness, and realism than VAE and GAN.

**Additional Feedback:**

The bottom line of Figure 1 (d) is okay for me and I can not find the counterfactual point.

**Documentation:**

Yes.

**Ethics:**

No.

**Limitations:**

Yes. The authors have discussed the limitation that some methods were left without discussion.

**Opportunities For Improvement:**

1. The image resolution in the study is very low, so some results (like beards) are hard to judge for human researchers. This may be important for study because although HVAE has advantages in counterfactual generation, it may not be able to generate high-quality images like GAN, auto-regressive models, and diffusion models.
2. The proposed method can not handle some recent representative generative paradigms like diffusion models so related experimental results are not provided. Though authors have claimed this in the limitation section.
3. The proposed study method strongly relies on learning a regressor (or a classifier) for each attribute, which would be very costly and hinder the study on datasets with massive categories or concepts.

**Relation To Prior Work:**

Yes.

**Summary And Contributions:**

Considering the importance of image realism in some fields such as natural and medical imaging, the authors of this work focus on the study of counterfactual image generation during image editing. They propose a framework based on Structural Causal Model (SCM) to check whether causal relationships are inherent in editing with the help of causal graphs and corresponding regressors and classifiers. Experiments on several datasets are conducted and various editing methods including VAE, HVAE, and GAN are benchmarked. The results of this study reveal the superiority of HVAE in composition, effectiveness, and realism.

---

> ### Author Rebuttal · Authors · 2024-08-17
>
> *The image resolution in the study is very low.*
>
> We agree with the reviewer that scaling to larger image resolutions would relate this work with the current state of non-causal image generation. However, we want to highlight that as we mention in the Introduction, unrestricted image generation/modification is a different task than causally faithful counterfactual image generation. We now elaborate the reasons we decided to evaluate on the current datasets and discuss our findings when aiming to scale the models to higher resolution.
>
> We included 3 types of image datasets: (a) synthetic [MorphoMNIST], (b) medical [ADNI], (c) real-world [CelebA].
> MorphoMNIST (a) is a dataset used by the majority of causal methods that work on images.
> ADNI (b) has a 192x192 resolution per slice as per the MRI acquisition protocol.
> CelebA (c) is widely used in the field and its set of attributes enables the community to devise causal graphs with a reasonable number of variables.
>
> We wanted to test the ability of our models to scale to resolutions that have not been designed for and for this reason we experimented with CelebA-HQ, an upscaled version of CelebA (of resolution 256x256).
> As can be seen on the Figure 2 of the general rebuttal pdf (qualitative results of CelebA-HQ with our best performing model: HVAE) the models failed to scale and enable a fair comparison on this resolution.
> The inability of VAEs and GANs to produce plausible results for large resolutions is well documented.
> For GANs different architectures such as StyleGAN, etc. have been developed to combat this issue, while HVAEs have been developed to combat this issue in the VAE family. However, even our HVAE model did not succeed on counterfactual generation, which can be attributed to the counterfactual training step needed to avoid the results of condition ignoring (see ln.\ 587 in Appendix A).
> This can be possibly interpreted by the fact that the gradients of the classifiers cannot accurately influence the model output in the pixel space as the dimensionality grows.
>
> Scaling the models would require a significant change on the architectures or conditioning of the models, which felt out of scope of the aim of this conference track. Nevertheless, if you see benefit, results can be included in the appendix.
>
> *The proposed method can not handle some recent representative generative paradigms like diffusion models*
>
> Please see general response.
>
> *The proposed study method strongly relies on learning a regressor (or a classifier) for each attribute, which would be very costly and hinder the study on datasets with massive categories or concepts.*
>
> The scalability of all benchmarked approaches (as well as most causal deep learning models) to graphs with large number of variables remains an open problem. Learning separate classifiers for each variable falls into this category. It is also a challenge for  image editing methods too (see e.g. classifier-guidance for diffusion models or the extra steps needed for classifier-free guidance).
> Our approach to move from synthetic, to natural to medical images, and from simple to more complex graphs, was proactively designed to approach real-world data problems, most of which are challenged by biases which correspond to relatively simple graphs. For example, is the "age", "gender", some "physiology factor" or a "response to therapy" that produces a change in an MRI image? Addressing the causal interplay of known biases and confounding factors in real world images is not trivial. Our approach aims to set the stage for a more thorough exploration on how adjusting known biases and confounding factors of increasing complexity, can improve realism in causal image editing.
>
> *The bottom line of Figure 1 (d) is okay for me and I can not find the counterfactual point.*
>
> We apologise if the point of Figure 1 is not evident and we didn't make it clear. Figure 1 (d) displays image edits that do not take into account the causal relationships of the data-generation process (non-causal image editing) to motivate the setup that we are benchmarking. Intervening on Age (\textit{do(Young)}) results in a young person with baldness which in most cases isn't realistic and is rather likely in the data.
> We used this example to motivate the reader of what happens if we don't generate causally faithful images.
> While it is possible for a young person to not have hair (e.g. due to early onset androgenic alopecia) our causal model allows for Age to influence Baldness not deterministically, but through a function that also takes noise as input allowing for stochasticity.

---

> ### Author Response · Authors · 2024-08-23
> **Follow up**
>
> Dear Reviewer HeXk,
>
> Thank you again for you hard work in reviewing this paper. We just wanted to follow up to see if you've had a chance to review and consider our response. Do let us know if it addresses your concerns, and please let us know if you have further suggestions or remaining concerns.

---

> ### Comment · Reviewer_HeXk · 2024-08-27
> **Thanks for the response**
>
> The authors' responses well address my concerns on Figure 1 (d). On the other hand, sufficient experimental results on diffusion models are provided to strengthen the generalization of the work, which is highly appreciated. However, I still have concerns about image resolution. In counterfactual image generation, though realness is important,  image quality is the basis for further usage (_e.g._, clear visualization is needed in medical imaging). Thus the scalability for larger resolution of the proposed method is of significance from my perspective (as well as the the dependence on the regressor / classifier).

---

> > ### Author Response · Authors · 2024-08-28
> >
> > We sincerely thank the reviewer for the positive message that we have addressed most of their comments regarding Figure 1d and diffusion models.
> >
> > We would like to clarify that the spatial resolution of the medical imaging data we used (ADNI1, 192x192 per slice) is imposed by the imaging protocol and is typical for tomographic imaging (MRI, CT, PET) in this case MRI. However, we do agree that it would be useful to be able to scale to larger spatial resolutions (higher spatial resolution in medical imaging when applicable, higher spatial resolution in biomedical imaging which involves much larger resolutions eg, cell imaging, in natural images, etc).
> >
> > At the moment we are bounded by the resolution and graph size abilities of the methods we benchmark. We are aware that improving image quality implies making methodological innovation. While we aim to explore novel conditioning mechanisms for diffusion models in future work, our purpose with this work is to provide the community with a thorough benchmarking framework (not propose new methodological innovations). As such we believe it is not within the scope of this paper, nor the Benchmarks and Datasets track of Neurips to make the required significant methodological changes to the underlying methods tested in order to increase their resolution capacity or the graph size (which would necessitate training more classifiers/regressors).
> >
> > In summary, we appreciate the reviewer's feedback and agree that higher resolution ability and scalability to larger causal graphs would be advantageous. However, we believe that addressing these aspects goes beyond the focus of this paper and the intended scope of this conference track which is to provide a thorough method comparison of existing generative model families and causal graphs and inspire future exciting work for the community to address using our evaluation framework. We aim to pursue these directions (scalability to higher resolution and conditioning diffusion models) in our future work.

---

> > > ### Comment · Reviewer_HeXk · 2024-08-29
> > > **Response from Reviewer HeXk**
> > >
> > > The efforts that the authors put into the explanation and clarification are appreciated. I agree that the greatest value of the work lies in benchmarking current art and inspiring future works and I have raised my rating.

---

> > > > ### Author Response · Authors · 2024-08-29
> > > >
> > > > We would like to thank again the reviewer for their hard work in reviewing our manuscript and the positive comments that we addressed all their concerns and feedback. We also sincerely thank the reviewer about expressing a revised score, after our rebuttal letter and interaction.

---

### Official Review · Reviewer_6W6G · 2024-07-24
**Thorough Design Choices, Relatively Limited Scope**

**Rating:** 7
**Confidence:** 3
**Correctness:** Yes.
**Clarity:** Yes.

**Review:**

The following points outline my thoughts regarding the submission.

1. **Presentation and Clarity**: With the exception of a few points, the paper is generally well-written and easy to follow. The authors do a good job of outlining the desiderata behind the assessment framework, considerations behind valid generative editing pipelines and the counterfactual intervention setup. My only gripe with the presentation is the lack of details in Section 3.2 and 3.3. The manuscript might benefit from importing some key details (how counterfactuals are exactly obtained; metrics for measurement) from the appendix into the main paper. For instance, I missed the fact that a subset of settings required fine-tuning specific to counterfactual generation.
2. **Originality**: As stated by the authors, the novelty of the proposed framework lies solely in comprehensive evaluation of counterfactual generation methods in terms of performance, fidelity and reliability. The underlying models, methods (with minor design choices) and datasets (with supporting counterfactual interventions) have been drawn from prior work. I think comprehensively evaluating along several such dimensions is valuable and informative.
3. **Quality and Significance**: I like the experimental setup adopted by the authors. They consider datasets of varying complexities (and from different domains) – helpful in outlining the effectiveness of a specific counterfactual editing approach across different data distributions. I also appreciate the thoroughness behind the choice of metrics (and supporting design choices – excluding reversibility), which do not require ground-truth counterfactuals. Overall, the experiments do provide a clear takeaway – HVAE outperforms other (considered) generative methods. However, this also leads me to a potential weakness – excluding diffusion models. While diffusion models may not be conditioned with non-trivial SCM graphs, we do have methods to (1) perform abduction [A] and (2) produce targeted edits (counterfactual [B,C] or otherwise [D, E]). Such (diffusion-based) mechanisms may not fall under the SCM framework, but excluding them substantially limits the scope of the proposed framework to very niche data domains.

[A] - Null-text inversion for editing real images using guided diffusion models, CVPR 2023

[B] - LANCE: Stress-testing Visual Models by Generating Language-guided Counterfactual Images, NeurIPS 2023

[C] - Instruct2Attack: Language-Guided Semantic Adversarial Attacks (https://arxiv.org/pdf/2311.15551)

[D] - Prompt-to-Prompt Image Editing with Cross Attention Control (https://arxiv.org/pdf/2208.01626)

[E] - Diffusion Model-Based Image Editing: A Survey (https://arxiv.org/pdf/2402.17525)

**Strengths:**

As discussed in the main review, I think the strengths of the submission lie in establishing a comprehensive framework to assess counterfactual image generation. I believe the framework on its own is potentially quite useful. Additionally, the authors do a good job of outlining the motivation, desiderata, design choices for and takeaways from such a framework.

**Additional Feedback:**

Please refer to the opportunites for improvement section.

**Documentation:**

Yes.

**Ethics:**

No.

**Limitations:**

Yes. The authors have discussed limitations associated with the proposed framework. Within the scope of experiments included in the paper, I don't see any major negative societal impact.

**Opportunities For Improvement:**

As stated in the main review, I think the primary weakness associated with the current submission is the exclusion of diffusion models. Even though they might not fit directly into the SCM framework, given that there are approximate techniques for abduction, excluding them substantially limits the scope of the proposed framework.

I think including experiments for those might improve the current submission. Perhaps along the lines of [B, C] -- to evaluate the quality of counterfactuals / adversarial samples generated by such mechanisms.

Other minor clarity issues can be addressed in a revision.

**Relation To Prior Work:**

Yes.

**Summary And Contributions:**

The paper presents a framework to compare different counterfactual image generation methods. Counterfactual image generation here refers to the realistic edits that respect causal factors that guide the data generation process. The authors consider several generative models that fit under the Structured Causal Model (SCM) framework – conditional, hierarchical VAEs, conditional GANs – and evaluate them across several dimensions – composition (reproducibility in absence of an intervention), effectiveness (success of intervention), realism (realistic edits) and minimality (localized changes in response to interventions). The authors assess counterfactual image generation abilities across multiple datasets (of increasing complexity) and note that hierarchical VAEs are consistently superior to other models across dimensions of evaluation. The authors provide supporting code and the assessment framework as a Python package to incentivize adoption of the proposed setup.

---

> ### Author Rebuttal · Authors · 2024-08-17
>
> *Lack of details in Section 3.2 and 3.3. The manuscript might benefit from importing some key details (how counterfactuals are exactly obtained; metrics for measurement) from the appendix into the main paper.*
>
> We agree with the reviewer that it is better for the metrics to be a part of the main paper, therefore we will augment Section 3.3 with the detailed descriptions of Appendix A.2 in the camera-ready.
> The same applies to the process of counterfactual training of the HVAE, for which we will move key details from Appendix A.1 to the main paper. We will also add more details on the second paragraph of Section 3.2 that describes how counterfactuals are obtained for all models.
> We will use the extra 1 page (to be provided), as well as extra space by merging tables 2-4 into one.
>
> *While diffusion models may not be conditioned with non-trivial SCM graphs, we do have methods to (1) perform abduction [A] and (2) produce targeted edits (counterfactual [B,C] or otherwise [D, E]). Such (diffusion-based) mechanisms may not fall under the SCM framework, but excluding them substantially limits the scope of the proposed framework to very niche data domains.*
>
> Please see general response on this matter.
>
> In addition, we want to thank the reviewer for providing a list of diffusion based models for image editing.
> We point, however, that the works mentioned perform text-controlled edits.
> Thus, to adapt them to our setting, fine-tuning with class conditioning is essential.
> An adaptation of the null-text inversion technique to perform more accurate noise abduction in our setting is a very interesting idea that we would be keen to explore as a part of a novel methodology for (non-trivial multi-variable) counterfactual image generation with Diffusion models.
> Another fundamental difference between the above image editing approaches and our causal SCM framework, is that the latter aims to unravel how the causal interplay of known high-level variables and confounding factors affect image realism and quality. In the proposed image editing approaches (e.g., see figures in the first 2 papers), there is commonly 1 prompt changing to produce an edit, which does produce realistic high-quality images, but does not reflect causal relationships among high-level variables. We assume that the latter is a statistically and computationally non-trivial but valid representation of the causal interplay between high-level variables.

---

> ### Author Response · Authors · 2024-08-23
> **Follow up**
>
> Dear Reviewer 6W6G,
>
> Thank you again for you hard work in reviewing this paper. We just wanted to follow up to see if you've had a chance to review and consider our response. Do let us know if it addresses your concerns, and please let us know if you have further suggestions or remaining concerns.

---

> > ### Comment · Reviewer_6W6G · 2024-08-25
> > **Thanks for the response!**
> >
> > Apologies for the delay in follow-up and thanks to the authors for sharing a detailed response to mine as well as other reviewers' concerns.
> >
> > I had 2 key points of improvement in my review - (1) improving details surrounding some design choices in the main paper and (2) incorporating diffusion models in the existing framework. (1) can be addressed easily and the authors promised to do so.
> >
> > Regarding (2), the authors provided an explanation of why it's hard to include diffusion models within the existing framework and also shared initial explorations in the general response. This is super-helpful and the manuscript would improve by including these explorations in the revised version. I would strongly encourage the authors to do so and also discuss the responses to concerns raised by other reviewers - especially experimental aspects like low-resolution and hindrances due to a regressor as well as other higher-level aspects (whether it makes sense to include non-causal methods).
> >
> > My concerns were sufficiently addressed in the revision and I am inclined to increase my rating of the submission.

---

> > > ### Author Response · Authors · 2024-08-29
> > >
> > > We would like to thank again the reviewer for their hard work in reviewing our manuscript and the positive comments that we addressed all their concerns and feedback. We also sincerely thank the reviewer about expressing a revised score, after our rebuttal letter and interaction.

---

### Official Review · Reviewer_J9ST · 2024-07-26
**Interesting and well-written benchmarking study**

**Rating:** 7
**Confidence:** 4
**Correctness:** Yes
**Clarity:** Yes

**Review:**

Interesting and well-written paper that addresses an important research gap and performs extensive benchmarking of existing methods. I have a few concerns about the applicability of the findings to more realistic datasets and methods, as well as the efficacy of some proposed metrics, but otherwise am positive about the paper’s overall contributions.

**Strengths:**

– The paper identifies and tackles an important research gap

– The paper is well-written and easy to follow

– The proposed datasets and metrics are intuitive

**Additional Feedback:**

None

**Documentation:**

Yes

**Ethics:**

No ethical issues

**Limitations:**

Yes

**Opportunities For Improvement:**

– The main limitation is the relatively few methods that are benchmarked, particularly the absence of denoising diffusion models. The paper reports noise abduction with such models as the primary challenge, but does not clarify why prior approaches that it cites [11, 40, 41] that treat noise abduction as forward diffusion are unsuitable. Further, is it not possible to perform counterfactual queries without noise abduction by using a simplified SCM (potentially at the cost of lower reliability)? In my opinion, the ability to benchmark diffusion models will significantly enhance the paper’s contributions. Otherwise, a detailed discussion of the challenges that make it infeasible would also be helpful for future work.

– Similarity, the paper only focuses on “true causal counterfactual methods” and exclude other semantic image editing methods (eg. EditGAN, StyleCLIP, StyleMapGAN etc.). It would be valuable to still benchmark the performance of (a subset of) such methods – do “true counterfactual methods” indeed outperform these methods on the proposed benchmark, as they should? If not, why so?

– The experiments are limited to simple datasets. The paper reports difficulties with scaling to higher-resolution datasets such as CelebHQ, but it would be helpful to explain these difficulties and why they are “somewhat expected” (L280). What specific aspects do the surveyed methods struggle with?

– The overall technical contribution is relatively limited: the paper benchmarks existing counterfactual image editing methods on existing datasets with a set of metrics proposed in prior work. That said, the paper’s main contribution: a unified framework (data, metrics, baselines) and codebase is still valuable to the community.

**Relation To Prior Work:**

Yes

**Summary And Contributions:**

Proposes a unified benchmark and accompanying codebase for counterfactual image editing within the structural causal model framework. Extensively tests the performance of three conditional image generation strategies on this benchmark along the axes of composition, effectiveness, minimality, and realism.

---

> ### Author Rebuttal · Authors · 2024-08-17
>
> *Why prior (diffusion) approaches are unsuitable?*
>
> Please see general response.
>
>
> *Why we can’t skip noise abduction using a simplified SCM?*
>
> Noise abduction is necessary for counterfactual generation under the Abduction-Action-Prediction paradigm (see 3.1, ln. 118). This paradigm is dominant in causality and even for non-interventional counterfactuals (such as backtracking [1]) noise abduction is still required. However, noise abduction can be performed in many ways and in the works we benchmark it takes the form of encoding into a latent space. Regarding the use of a simplified SCM, as stated in the general comment, previous works would restrict us to using only a single-variable SCM which is a trivial causal setting.
>
>
> *It would be valuable to still benchmark the performance of non-causal methods.*
>
> Indeed, at first it would seem that such comparison is worthwhile, but it will not be comparing apples to apples. As we mention in the Introduction (ln. 23) causal counterfactuals add more restrictions. Non-causal methods reside on the better explored and theoretically relaxed field of image editing.
> There is broad acceptance in the field that counterfactual image generation quality is not up to  par compared to non-causal methods.
> Our paper hence focuses on providing a framework to benchmark and evaluate causal counterfactual image generation methods. We provide examples of non-causal ones to motivate the need for being causally faithful (see Figure 1).
>
> Note that the suggested image-editing methods (eg. EditGAN, StyleCLIP, StyleMapGAN etc.) require either mask or text description which are not only incompatible with our setup but also increase the amount of information provided to the system.
> Once causal counterfactual image generation methods advance (in image resolution capability) and graph size to handle such additional inputs we agree fully that a comparison between causal and non-causal methods will be interesting. We still believe that on its own, comparing causal counterfactual methods against each other in a common framework and footing is still a valuable contribution for the field.
>
>
> *The experiments are limited to simple datasets. difficulties with scaling to higher-resolution datasets such as CelebHQ. What specific aspects do the surveyed methods struggle with?*
>
> We agree with the reviewer that scaling to larger image resolutions would relate this work with the current state of non-causal image generation. However, we want to highlight that as we mention in the Introduction, unrestricted image generation/modification is a different task than causally faithful counterfactual image generation. We now elaborate the reasons we decided to evaluate on the current datasets and discuss our findings when aiming to scale the models to higher resolution.
>
> We included 3 types of image datasets: (a) synthetic [MorphoMNIST], (b) medical [ADNI], (c) real-world [CelebA].
> MorphoMNIST (a) is a dataset used by the majority of causal methods that work on images.
> ADNI (b) has a 192x192 resolution per slice as per the MRI acquisition protocol.
> CelebA (c) is widely used in the field and its set of attributes enables the community to devise causal graphs with a reasonable number of variables.
>
> We wanted to test the ability of our models to scale to resolutions that have not been designed for and for this reason we experimented with CelebA-HQ, an upscaled version of CelebA (of resolution 256x256).
> As can be seen on the Figure 2 of the general rebuttal pdf (qualitative results of CelebA-HQ with our best performing model: HVAE) the models failed to scale and enable a fair comparison on this resolution.
> The inability of VAEs and GANs to produce plausible results for large resolutions is well documented.
> For GANs different architectures such as StyleGAN, etc. have been developed to combat this issue, while HVAEs have been developed to combat this issue in the VAE family. However, even our HVAE model did not succeed on counterfactual generation, which can be attributed to the counterfactual training step needed to avoid the results of condition ignoring (see ln.\ 587 in Appendix A).
> This can be possibly interpreted by the fact that the gradients of the classifiers cannot accurately influence the model output in the pixel space as the dimensionality grows.
>
> Scaling the models would require a significant change on the architectures or conditioning of the models, which felt out of scope of the aim of this conference track. Nevertheless, if you see benefit, results can be included in the appendix.
>
>
> *The overall technical contribution is relatively limited.*
>
> Hence, we do find the exploration of diffusion models and conditioning mechanisms to be highly intriguing but we are keen to keep this in our future work. Our paper fits the scope of the track which is that of benchmarking. Its novel contribution does not reside on introducing new methods, but on introducing a systematic framework and comparison in terms of different combinations of model families, datasets and causal graphs across metrics. In our quest delivering the above we also had to introduce methodological innovations to extend methods to untested datasets and causal graphs (see ln. 51-53 and the Setup paragraph in Section 4).
>
> [1] Kugelgen, J.V., Mohamed, A., Beckers, S.: Backtracking counterfactuals. PMLR 2023

---

> ### Author Response · Authors · 2024-08-23
> **Follow up**
>
> Dear Reviewer J9ST,
>
> Thank you again for you hard work in reviewing this paper. We just wanted to follow up to see if you've had a chance to review and consider our response. Do let us know if it addresses your concerns, and please let us know if you have further suggestions or remaining concerns.

---

> > ### Comment · Reviewer_J9ST · 2024-08-28
> > **Rebuttal addresses concerns**
> >
> > I am satisfied by the author response, particularly the additional experiments with diffusion models and the clear justification behind not benchmarking image editing methods. I will raise my rating.

---

> > > ### Author Response · Authors · 2024-08-29
> > >
> > > We would like to thank again the reviewer for their hard work in reviewing our manuscript and the positive comments that we addressed all their concerns and feedback. We also sincerely thank the reviewer about expressing a revised score, after our rebuttal letter and interaction.

---

### Author Rebuttal · Authors · 2024-08-17

We thank all reviewers for their really valuable and helpful feedback! In the general rebuttal we address remarks regarding diffusion models. In the reviewer specific comments we focus on addressing the rest of the reviewers' suggestions and comments

We appreciate the suggestion to include diffusion models in this work, given that they constitute a (recently) successful generative model family. Our original intention was to incorporate diffusion models into our experimental framework. However, at the time of submission, we were unable to successfully adapt existing methodologies (e.g. [1-3]) to accommodate the multi-variable context necessary for representing causal interplay and confounding variables. The published methods were primarily designed for single-variable scenarios, posing a significant challenge for their direct application to our research problem. Therefore, we considered that this is necessitating the investigation of a novel diffusion model method, to accommodate multiple variables.
As our primary objective was to align our benchmarking efforts with the specific scope of this track, we refrained from incorporating such methodological innovations in the previous version of our submission. We were concerned that these innovations might be perceived as falling outside the defined scope of the track.

Nevertheless seeing your comments, we now discuss our investigations on this matter. We will also expand discussion on diffusion models in the manuscript.

To provide more context, we now describe the diffusion models that we developed. We extended the classifier-free guidance setting of [1] in graph sizes larger than the original paper on the benchmarking datasets.
Initially this method under-performed (which we confirmed with the authors as a limitation of their model) when modeling non-trivial graphs (with multiple variables), which can be partially attributed to the spurious correlation of the variables.
Also, we empirically observed that the nature of the conditioning mechanisms used and the architectural and hyperparameter choices utilised, can make an important difference for the performance of diffusion models. We incorporated this model (UNet2DConditionModel taken from the diffusers library [4]) in our codebase and tested it using our framework.
We followed the same setting as in all our models, treating the forward diffusion process as the noise abduction and the backward process as the prediction.

During the rebuttal period (and due to the large training and evaluating times that these models required), we managed to run experiments on MorphoMNIST, as well as the CelebA (simple) datasets. We present the qualitative and quantitative results in the rebuttal pdf.
We report that for MorphoMNIST, the Diffusion model performs worse than the rest of the models in terms of composition and realism (FID), while its performance on effectiveness and minimality (CLD) is comparable to that of the GAN. This is somehow expected, as this dataset is not complex in terms of image content and our simpler models, such as VAE perform well.
For CelebA (using the simple causal graph), the diffusion model performs excellent on composition for 1 cycle, but it deteriorates significantly after 10 cycles. Regarding effectiveness, its performance is slightly worse than the other models, but ranks second when measuring its effect of intervening on eyeglasses.
Finally, the realism and minimality of generated images is fairly good but worse than the HVAE.
We assume that the underperformance of the current Diffusion model in the task we are benchmarking is related to the conditioning mechanism we incorporated. As this extension is a naive approach, we believe that novel methodological contribution is needed to enable a more fair comparison with other generative models that have been already proposed for counterfactual image generation in this setting.

We suggest that, since there is no published work that explores the use of diffusion models in the setting we benchmark, to include major observations and difficulties around the use of diffusion models for counterfactual generation in the discussion. This would be the most useful for the community but we are keen to have your thoughts.

[1] Sanchez, P., Kascenas, A., Liu, X., O’Neil, A., Tsaftaris, S.: What is healthy? generative counterfactual diffusion for lesion localization. In: Mukhopadhyay, A., Oksuz, I., Engelhardt, S., Zhu, D., Yuan, Y. (eds.) Deep Generative Models - 2nd MICCAI Workshop

[2] Sanchez, P., Tsaftaris, S.A.: Diffusion causal models for counterfactual estimation. In: CLEaR (2022)

[3] Fontanella, A., Mair, G., Wardlaw, J., Trucco, E., Storkey, A.: Diffusion models for counterfactual generation and anomaly detection in brain images (arXiv:2308.02062) (Aug 2023)

[4] von Platen, P., Patil, S., Lozhkov, A., Cuenca, P., Lambert, N., Rasul, K., Davaadorj, M., Nair, D., Paul, S., Berman, W., Xu, Y., Liu, S., Wolf, T.: Diffusers: State-of-the-art diffusion models. https://github.com/huggingface/diffusers (2022)

---

### Decision · Program_Chairs · 2024-09-26

**Decision:**

Accept (Poster)

**Comment:**

All reviewers unanimously agreed that this paper is worthwhile to be presented at NeurIPS. Although there have been several concerns in the initial review, the authors provided prompt and useful additional information and explanation during the rebuttal. The authors are encouraged to reflect the reviewers' comments in the final version. Also, it would be interesting to extend this work to incorporate diffusion models as a future work.